# Facultative dosage compensation of developmental genes on autosomes in Drosophila and mouse embryonic stem cells

Claudia Isabelle Keller Valsecchi [1], M. Felicia Basilicata [1], Giuseppe Semplicio[1], Plamen Georgiev[1], Noel Marie Gutierrez[1] & Asifa Akhtar [1]

Haploinsufficiency and aneuploidy are two phenomena, where gene dosage alterations cause severe defects ultimately resulting in developmental failures and disease. One remarkable exception is the X chromosome, where copy number differences between sexes are buffered by dosage compensation systems. In *Drosophila*, the Male-Specific Lethal complex (MSLc) mediates upregulation of the single male X chromosome. The evolutionary origin and conservation of this process orchestrated by MSL2, the only male-specific protein within the fly MSLc, have remained unclear. Here, we report that MSL2, in addition to regulating the X chromosome, targets autosomal genes involved in patterning and morphogenesis. Precise regulation of these genes by MSL2 is required for proper development. This set of dosage-sensitive genes maintains such regulation during evolution, as MSL2 binds and similarly regulates mouse orthologues via Histone H4 lysine 16 acetylation. We propose that this gene-by-gene dosage compensation mechanism was co-opted during evolution for chromosome-wide regulation of the *Drosophila* male X.

---

[1] Max Planck Institute of Immunobiology and Epigenetics, Stübeweg 51, 79108 Freiburg, Germany. These authors contributed equally: M. Felicia Basilicata, Giuseppe Semplicio.  Correspondence and requests for materials should be addressed to A.A. (email: akhtar@ie-freiburg.mpg.de)

Alteration of gene dosage including whole chromosome gain and loss has a profound impact on cellular physiology and is characteristic of disease states, for example, in human cancer cells. Sex chromosomes represent an interesting exception where aneuploidy is naturally tolerated, most probably due to the action of dosage compensation (DC) mechanisms[1]. Since its initial discovery in *Drosophila*, DC has gained particular attention as a prototype example of a chromatin-linked process that is orchestrated by non-coding RNAs (ncRNAs) in various organisms including mammals[2–5].

In flies, DC manifests in males and its deficiency results in male-specific lethality predominantly at the 3rd instar larval (L3) stage[6]. DC is mediated by the male-specific lethal complex (MSLc), which consists of the proteins MSL1, MSL2, MSL3, MOF, and two functionally redundant ncRNAs roX1 and roX2. Incorporation of those into the core MSLc requires the activity of the RNA-associated RNA helicase MLE. The MSL complex binds the male X chromosome, where MOF catalyzes Histone H4 lysine 16 acetylation (H4K16ac) ultimately leading to an upregulated gene expression[7,8]. However, most of the MSL-complex members are expressed and regulate biological processes in both sexes[9–13]. Thus, DC-specific functions of the MSLc most likely arise from MSL2 and/or roX ncRNAs, which are exclusively expressed in males. Biochemical, structural, and tissue culture studies have revealed that MSL2 is the key determinant in MSLc targeting to X-linked DNA elements called High-Affinity Sites (HAS)[14–16]. However, the actual set of X-chromosomal genes requiring such regulation in vivo has remained elusive. Given the relatively late lethality of *msl* loss-of-function mutants at the L3 stage, it is unlikely that these genes function in the general maintenance of cellular physiology, but presumably rather modulate the activation of developmental pathways. A combination of subtle changes resulting in specific responses upon a haploinsufficient state in *msl*-mutant males may thereby ultimately result in lethality[17]. Moreover, it is conceivable that the full repertoire of MSL target genes is much more diverse and dynamic in different cellular contexts, a scenario that applies, for example, to mammalian MSLs[18,19]. We therefore envisioned that it is crucial to explore the roles of the MSL2 in vivo and for this, developed a highly optimized procedure for ChIP and CLIP/FLASH[20]. Unexpectedly, we discover that MSL2 not only binds to the X chromosome but also to autosomal promoters involved in patterning and morphogenesis. We show that the precise regulation of these genes by MSL2 is required for proper wing and eye development. Moreover, we find that this gene-by-gene dosage compensation mechanism is conserved in mammalian cells and represents a presumably ancient function of the core MSLc. Taken together, we propose that the regulatory role of MSL2 extends beyond X-linked genes and involves regulation of dosage-sensitive genes on autosomes.

## Results

### The MSL-mediated chromatin landscape in Drosophila larvae.
In order to better understand the regulatory networks orchestrated by the MSLc members in vivo, we decided to comprehensively explore the DNA and RNA interaction network of MSL2 and MLE in *Drosophila* L3 larvae. Therefore, we developed an optimized procedure to extract nuclear and chromatin-associated proteins for generating high-resolution DNA (Micrococcal Nuclease (MNase) ChIP-seq) and RNA (CLIP/FLASH) interaction profiles (Supplementary Data 1). For MSL2 ChIP-seq we generated a *UAS-msl-2::3Flag*-tagged transgene, which was expressed using a *tub-Gal4* driver to match endogenous MSL2 levels and accordingly rescued lethality as well as gene expression defects of *msl-2* null mutant males. Expression of *UAS-msl-*

*2::3Flag* also resulted in faithful localization of the MSLc members in polytene squashes (Supplementary Fig. 1a–e). As this transgene, henceforth referred to as MSL2tg, lacks UTRs and introns required for SXL-dependent repression of MSL2 expression[21], we were also able to assess MSLc targeting in females ectopically expressing MSL2tg.

The high-resolution inherent to MNase ChIP-seq (see extensive discussion in ref. [22]) led us to analyze small (≤140 bp) and large (>140 bp) reads separately. In brief, small fragments represent the actual footprint of a given DNA-binding protein, whereas large fragments reflect crosslinking events with adjacent nucleosomes and/or indirect contacts (Supplementary Fig. 1f). We instantly spotted that the MSL2tg profiles in males and females looked essentially identical with an excellent correlation on the merged set of all called peaks versus the untagged controls (Supplementary Fig. 1g). We conclude that ectopic expression of MSL2tg in females is sufficient to trigger proper targeting at molecular level in vivo. As expected, MLE and MSL2tg peaks were found at X-chromosomal promoters and HAS, for example, *Pp2C1* and *roX1* (Fig. 1a–c, Fisher's exact test for significant overrepresentation of X-linked peaks in MSL2tg <2.2e-16). Surprisingly though, MSL2tg but not MLE peaks were also found on autosomes and displayed similar enrichment levels compared to the X (Fig. 1a, b). Amongst the 1684 autosomal peaks, 970 were found within 200 bp of a transcription start site (TSS) indicating that a major fraction of the autosomal binding events occurs on promoters (see below).

We next compared the chromatin landscape at the X-linked and autosomal MSL2tg binding sites in further detail. On the X, we found MSL2tg enrichment at HAS centers, from which spreading and engagement with neighboring nucleosomes was detectable in the reads larger than 140 bp (Fig. 1c). The in vivo binding of MSL2tg at HAS is in great agreement with MSL2 profiles generated in S2 cells[23] and correlates with MSL1 enrichment in salivary glands[10]. HAS were also enriched for roX[24] and MLE, the latter of which in our high-resolution profiles does not enrich at the center, but at the neighboring nucleosomes surrounding HAS. Moreover, these nucleosomes appear well-positioned and H4K16-acetylated in males but not females, whereas other H3 and H4 acetylation marks are absent (Fig. 1d). To address, whether H4K16ac at these sites is MSL-mediated, we generated profiles of H4K16ac in *msl-2*Δ flies, in which the entire coding sequence had been deleted using a CRISPR/Cas9-mediated approach (Supplementary Fig. 1h). Analysis of the input coverage confirmed the presence of a single X chromosome in male samples (Supplementary Fig. 1i). The H4K16ac ChIP-seq profile upon deletion of *msl-2* in males looked essentially identical to the one in wild-type females. The enrichment of H4K16ac, as well as the well-positioned nucleosome organization at HAS vanished (Fig. 1d). Moreover, H4K16ac was collectively lost on male X-linked gene bodies, whereas TSS-proximal H4K16ac on both X and autosomes remained unaffected (Fig. 1e). The latter are characterized by high levels of H3K36me3 towards the TES, H3ac and H4ac near the TSS, and a marked enrichment of NSL3 precisely between those hyperacetylated nucleosomes, implying that H4K16ac at these sites is catalyzed by MOF residing in the NSL complex (Supplementary Fig. 1j)[25]. Taken together, these data suggest that the MSL-mediated chromatin environment at HAS is largely comparable between L3 larvae and S2 cells.

### MSL2 binding at autosomal genes.
Having established that our in vivo profiles of MSL2tg recapitulate binding on the X (Fig. 1 and Supplementary Fig. 2a), we next explored the autosomal sites. Upon manual inspection of the MSL2tg peaks in the genome browser, we noticed that autosomal peaks were frequently found

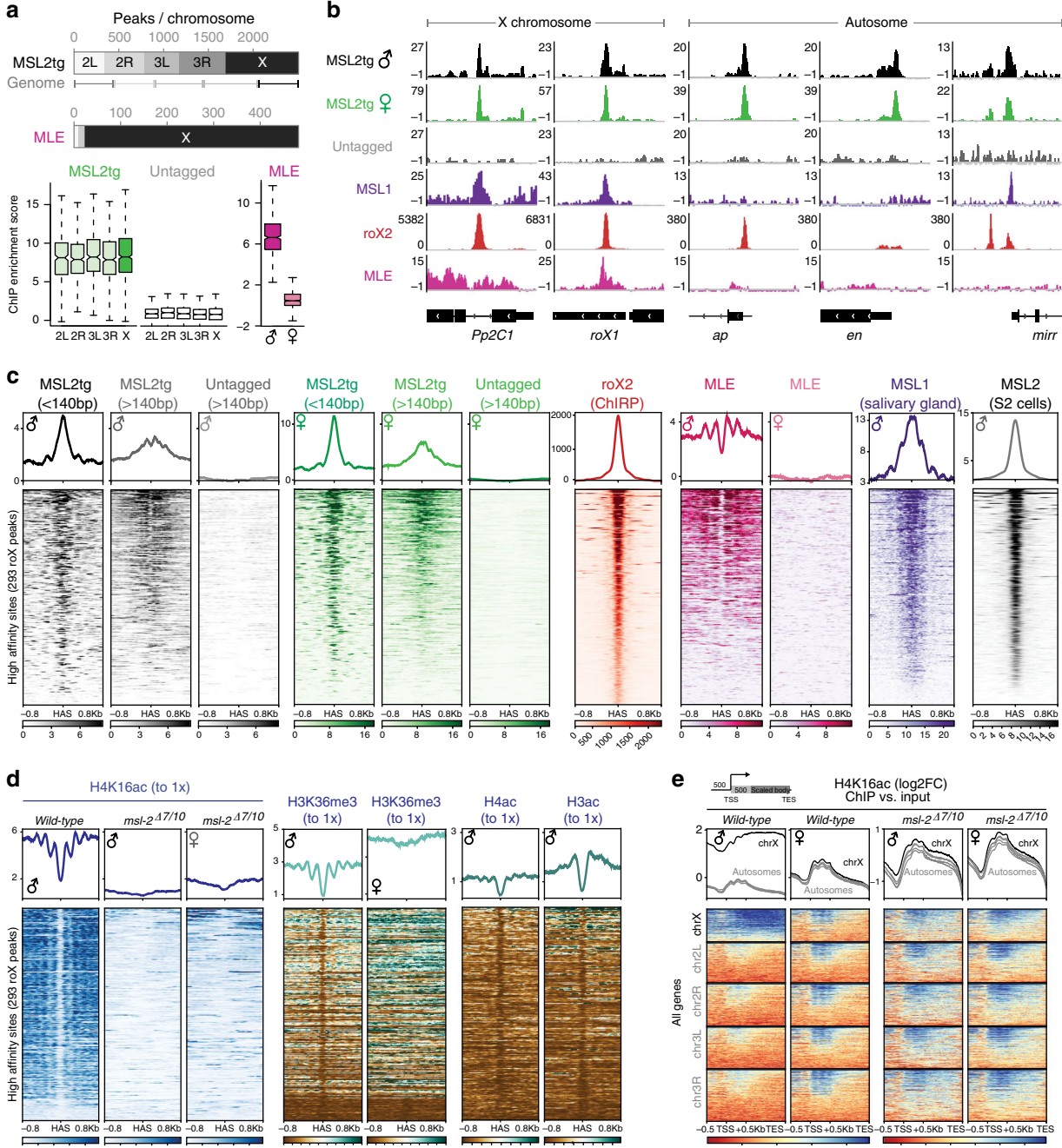

**Fig. 1** High-resolution profiling of the MSL-mediated chromatin landscape on the *Drosophila* X chromosome. **a** Scheme illustrating the total number of MACS2 peaks per chromosome collectively called in MSL2tg or MLE ChIP-seq samples and replicates. The line below the scheme indicates the size of the corresponding chromosomal arms. The *P*-value (Fisher's exact test) for significant overrepresentation of X-linked peaks in MSL2tg is <2.2e−16. The boxplots display the ChIP enrichment intensity on all MACS2 peaks per chromosome. Enrichment scores were calculated using deepTools multibigwigsummary. **b** Genome-browser snapshot of the X-linked *Pp2C1* (4C12-4C13) and *roX1* (3F3-3F3) and the three autosomal regions *ap* (41F-41F8), *en* (47F17-48A1), and *mirr* (69D3-69D4). MSL1 ChIP was from ref. [10], roX2 ChIRP from ref. [24] Note the lower ChIRP enrichment levels of roX2 at autosomal sites in comparison with X-linked HAS. **c** Heatmaps showing the normalized ChIP enrichment on HAS sorted by enrichment intensity. The HAS center was used as a reference point, while plotting the signal ±0.8Kb. The MSL2tg ChIP-seq reads were split according to their fragment length. The mean enrichment profile is shown on top of the heatmap, ChIP data normalization is described in methods. MLE ChIP profiles were generated from wild-type Oregon R, MSL2tg from *msl-2²²⁷/msl-2^km*, *tub-Gal4/UAS-msl-2::3Flag* L3 larvae, where only transgenic, but not endogenous MSL2 is expressed. MSL1 ChIP was from ref. [10], roX2 ChIRP and HAS definition from ref. [24], MSL2 ChIP from ref. [23]. **d** As in **c**, Heatmap showing the ChIP enrichment on HAS in wild-type and *msl-2^Δ7/msl-2^Δ10* L3 larvae. **e** As in **c**, Heatmap showing the ChIP enrichment on all annotated genes split per chromosome and sorted within each group according to the enrichment intensity. The transcription start site (TSS) was used as a reference point for plotting, while leaving ±0.5Kb unscaled. The rest of the gene body until the transcription end site (TES) was scaled to 1 Kb. The mean enrichment profile is shown on top (autosomes gray, X chromosome black)

at promoters of developmental regulatory genes, for example, *apterous* (*ap*), *engrailed* (*en*), or *mirror* (*mirr*) (Fig. 1b). Importantly, they were absent in untagged controls, excluding that they are false positive phantom peaks (Figs. 1b and 2a)[26]. We compared the chromatin landscape at these sites with HAS using an unsupervised clustering approach and noticed some differences. First, a subset of autosomal MSL2tg peaks (Cluster 1) showed

enrichment for roX and the adaptor protein CLAMP[24,27] (Fig. 2a). Although the CLAMP ChIP enrichments on autosomal Cluster 1 peaks were comparable to the ones found at HAS (Supplementary Fig. 2b), roX accumulated at substantially lower levels. Second, only modest levels of H4K16ac can be found in the regions surrounding these sites. Third, we do not detect MLE at or around at these peaks. MSL2 binding to a subset of HAS occurs

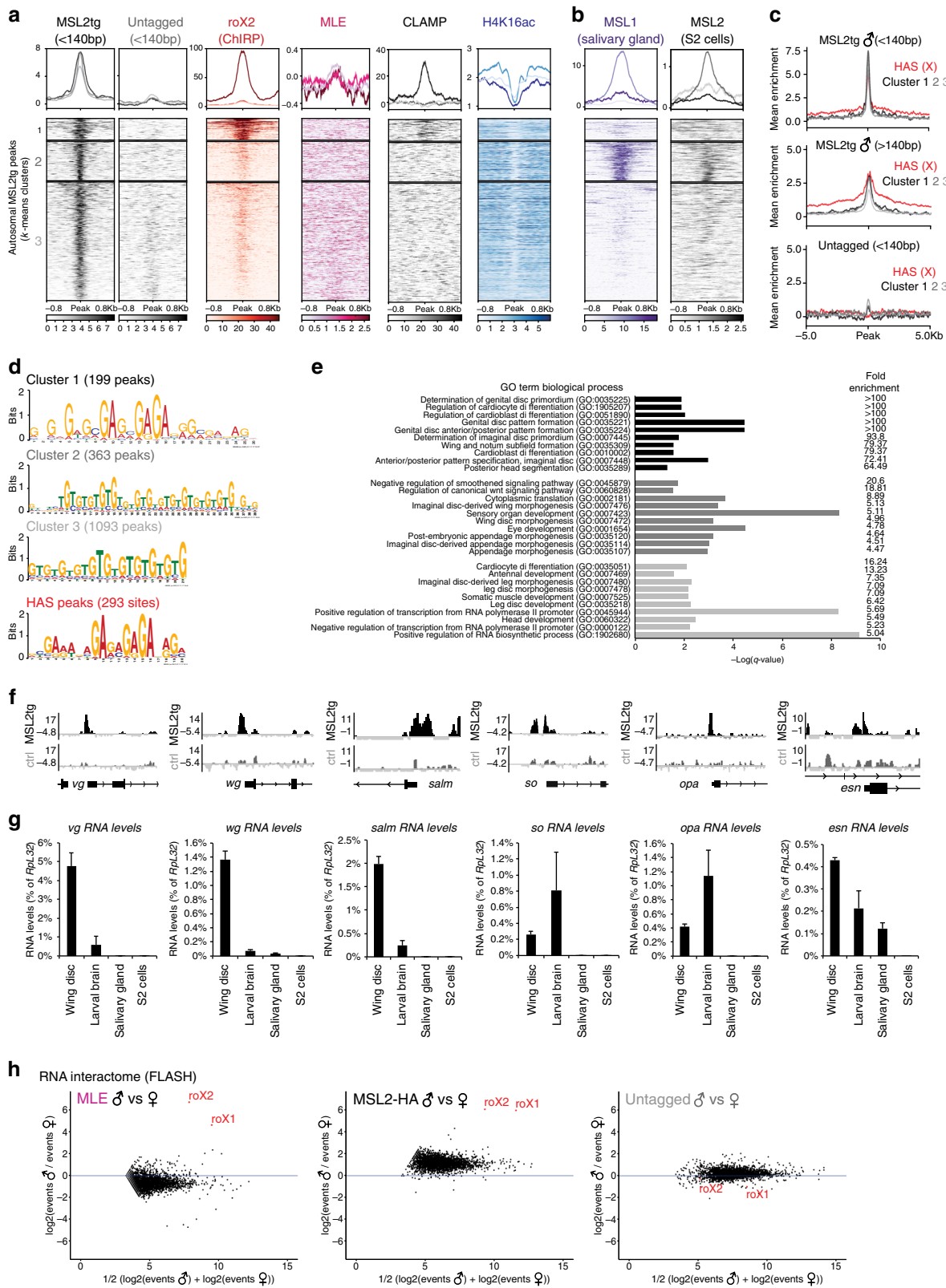

in the absence of MLE[15] and MLE was suggested to be required for spreading of MSL2[28]. In agreement with this, the local binding of MSL2tg on the autosomal sites is as pronounced as on HAS, but spreading is not observed (Fig. 2c). This contrasts the situation on the X, where MSL2tg and MLE spreading can be detected on the Kb scale within H3K36me3-positive domains, while H4K16ac spans more broadly (Supplementary Fig. 2c). Fourth, these sites are not MSL2-bound in S2 cells[23] and only a subset (Cluster 2) shows MSL1 enrichment in salivary glands[10] (Fig. 2b) pointing towards a tissue-specificity of MSL2 binding on autosomes.

Given the role of the MSL2 CXC domain in direct DNA recognition[29], we next performed motif analyses using MEME (Fig. 2d). In agreement with the enrichment of CLAMP[27], Cluster 1 peaks display a motif that is very similar to the well-known GAGA-rich sequence found at HAS[16], whereas Cluster 2/3 sites are characterized of a TG-rich sequence[15,30]. The TG-rich motif was also found at X-linked promoters (Supplementary Fig. 2d), suggesting that similar DNA recognition mechanisms might be in place for MSL2 recruitment on both X and autosomes.

Next, we were interested to address whether genes displaying autosomal MSL2tg promoter peaks function in a certain biological pathway and performed GO term analyses (Supplementary Data 2 and Fig. 2e). Consistent with the absence of those peaks in S2 cells, MSL2tg-bound genes function in developmental processes, for example, cell differentiation, regulation of the Wnt and Smoothened-signaling pathways or morphogenesis. Among the autosomal targets that were promoter bound by MSL2tg were well-known developmental regulatory genes, for example, *vestigial* (*vg*), *wingless* (*wg*), or *sine oculis* (*so*)[31] (Figs. 1b and 2f). Such genes were neither expressed in S2 cells nor in salivary glands (Fig. 2g and Supplementary Fig. 2e), the two systems where MSL2 targeting has been previously studied using ChIP-seq[23,32]. Moreover, these genes were not misregulated in *mle* null mutant females indicating that they do not functionally overlap with autosomal MLE binding sites detected in female salivary glands[33]. To exclude that this binding is not a mere artifact caused by using a *UAS-msl-2* transgene for ChIP-seq, we generated flies, where the endogenous *msl-2* gene is C-terminally tagged using CRISPR/Cas9 (Supplementary Fig. 2g–j). We performed ChIP-qPCR experiments and confirmed the binding of MSL2-HA to *vg*, *ap*, and *en* at levels comparable to HAS, whilst two autosomal controls (*ent2* and *CG15011* promoter) were not enriched (Supplementary Fig. 2i). We also investigated, whether at the RNA level the in vivo interactome of MSL2 and MLE in L3 larvae would be equally diverse and performed FLASH analyses to detect RNA–protein interactions[20]. However, our experiments revealed that the major RNAs bound by both MLE and MSL2-HA are indeed roX1 and roX2 (Fig. 2h and Supplementary Fig. 2k). Collectively, our ChIP data revealed that the repertoire of MSL2 binding sites is much more diverse in vivo than anticipated from tissue culture cells. This pointed towards the possibility, that MSL2 may be involved in the regulation of dosage-sensitive genes on autosomes, which is mechanistically distinct from the chromosome-wide regulation of the X.

**MSL2 regulates developmental patterning genes on autosomes.** Our ChIP profiles represent an average over all the tissues present in the anterior part of a L3 larva, which includes imaginal discs forming, for example, wings, eyes, or antennae. MSL2tg-bound genes such as *ap* or *wg* are well-known for regulating developmental pathways during wing morphogenesis. Therefore, we decided to use wing imaginal discs for exploring, whether MSL2 regulates such patterning genes on autosomes. Taking into consideration that MSL2 is only expressed in male flies, we firstly assayed sex-specific expression differences using qPCR. As expected, the X-linked *roX1* and *roX2* transcripts were male-specifically expressed, whereas *Klp3a* and *Ucp4a* appeared fully dosage compensated (Supplementary Fig. 3a). Among the selected autosomal MSL2 target genes, *so* displayed a significantly higher male expression level in these bulk analyses of an entire wing disc (Fig. 3a). We also assayed two prominent X-linked genes involved in wing morphogenesis, *Notch* (*N*) and *Beadex* (*Bx*), and were surprised to find that they were significantly less expressed in males and hence, escape DC (Fig. 3a and Supplementary Fig. 3b). In agreement with this, both genes did not show substantial enrichment for H4K16ac in their gene body, while this could be readily detected on dosage-compensated genes such as *Ucp4a* or *Klp3a* (Supplementary Fig. 3c). Given that *N* is haploinsufficient[34], sensitivity to dosage alterations within the Notch signaling pathway might be different in males versus females.

We therefore set out to test, whether modulating *msl-2* affects expression levels of target genes on both X and autosomes. Heterozygous *msl-2* mutants display equal MSL2 protein levels compared to wild-type males (Supplementary Fig. 1a) and *msl-2^RNAi* using *tub-Gal4* resulted in male-specific (35390) or non-specific (31627) lethality (Fig. 4d). Therefore, we decided to perform the inverse experiment and assess expression changes upon ectopic expression of MSL2tg in females (Fig. 3b, Supplementary Fig. 3d–f). Indeed, we observed an upregulation of the autosomal MSL2 target genes *vg*, *ap*, *wg*, and *so*. Among the X-linked genes, *roX2* and *Klp3a* were upregulated, whereas *Ucp4a, CG5254, N, Bx*, and several autosomal controls, as well as the same set of genes in males remained unaffected

**Fig. 2** MSL2 binding on autosomes. **a** Heatmaps showing the normalized ChIP enrichment on MSL2tg autosomal peaks. Three unsupervised *k*-means clusters were generated and the signal sorted by enrichment intensity within each cluster. The peak center was used as a reference point, while plotting the signal ±0.8Kb. The mean enrichment profile is shown on top of the heatmap, ChIP data normalization is described in methods. MLE and H4K16ac ChIP profiles were generated from wild-type Oregon R L3 larvae, MSL2tg ChIPs from transgenic *msl-2^227/msl-2^km, tub-Gal4/UAS-msl-2::3Flag* male L3 larvae. roX2 ChIRP was from ref. [24], CLAMP ChIP from ref. [27]. **b** As in **a**, plotting the MSL1 (salivary glands) ChIP from ref. [10] and the MSL2 (S2 cells) ChIP from ref. [23]. **c** Mean MSL2tg ChIP enrichment profile on HAS (red) compared to autosomal peaks in Clusters 1–3 (black/gray). **d** MEME motif analysis on MSL2tg peaks. The top-scoring motif was chosen for each autosomal peaks cluster and HAS. **e** GO term analysis (biological process) of the genes in each cluster associated with an MSL2tg peak. Genes associated with MSL2tg peaks were defined as overlapping within 200 bp of the TSS. **f** Genome-browser snapshots of the selected autosomal regions *vg* (49E1-49E1), *wg* (27F1-27F1), *salm* (32F1-32F2), *so* (43B3-43C1), *opa* (82D8-82E1), and *esn* (42F1-42F1) showing MSL2tg ChIP-seq enrichment in males with corresponding untagged controls. **g** Real-time RT-qPCR analyses of the indicated genes in wild-type Oregon R male wing discs, L3 larval brains, salivary glands or S2 cells. The RNA level of each gene was calculated relative to *RpL32* expression as a reference gene. The barplot represents the average of 3–5 independent biological replicates with error bars indicating the SEM. **h** FLASH experiments from separated male and female wild-type Oregon R or *msl-2::HA* L3 larvae. MA-plots show X-linking event counts at the gene level for pairwise biological replicates of MLE, MSL2, and untagged control (single replicate) between males and females

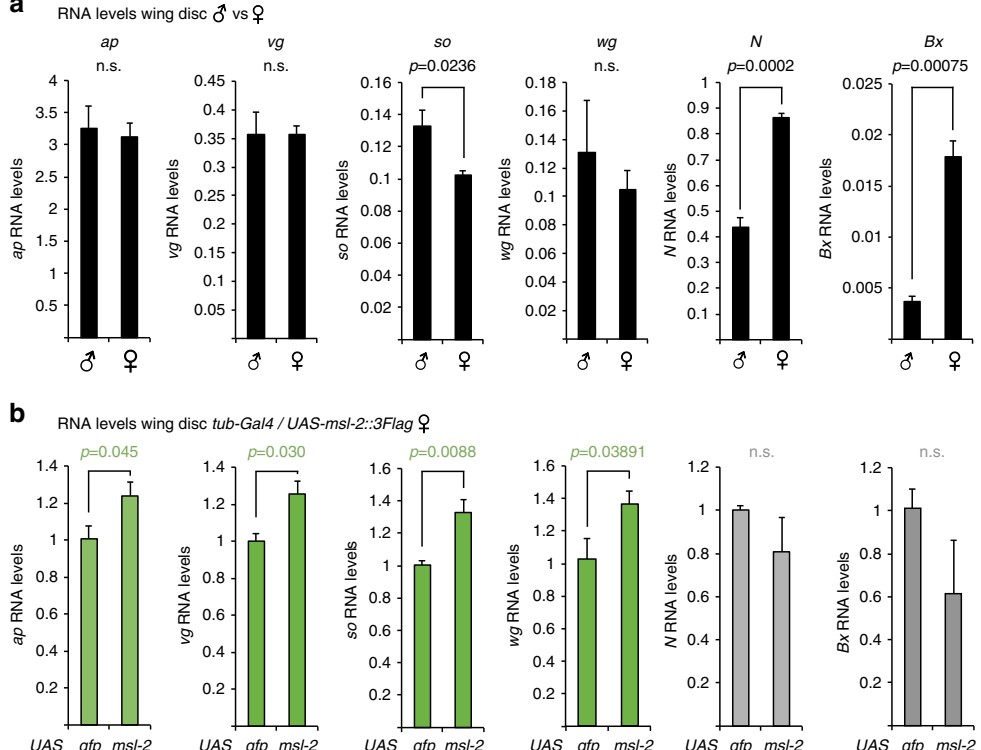

**Fig. 3** MSL2 regulates developmental patterning genes on autosomes. **a** Real-time RT-qPCR analyses of the indicated genes in wild-type Oregon R male and female wing discs. The RNA level of each gene was calculated relative to the geometric mean of *RpL32*, *Pfk*, and *U6* expression level. The barplot represents the average of four independently collected samples each consisting of two wing discs with error bars indicating the SEM. *P*-values were calculated using a one-tailed *t*-test (males versus females). **b** As in **a**, the data are expressed relative to the *UAS-GFP* expressing control female wing discs. *P*-values were calculated using a one-tailed *t*-test (*UAS-msl-2::Flag* flies versus *UAS-gfp* controls). The genotype of the flies was w;; tub-Gal4/UAS-msl-2::3Flag or w;; tub-Gal4/UAS-gfp

(Supplementary Fig. 3f, g). Immunofluorescence (IF) stainings performed on female wing discs revealed that MSL2tg does not localize to a typical H4K16ac-positive territory characteristic for the male X (Supplementary Figs. 3d, e and 4a, b and e). A subclass of HAS are bound by MSL2 in the absence of MLE and are proposed to be the first binding sites upon initiation of DC[15]. In light of these data, our findings suggest that in this intermediate state in females, MSL2 may have a more acute regulatory role on both autosomal sites (e.g., *ap*, *wg*, *vg*, *so*) and MLE-independent HAS (e.g., *roX2*, *Klp3a*) that is distinct from the chromosome-wide, spreading-dependent regulation of the X chromosome (e.g., *Ucp4a*, *CG5254*).

**MSL2 depletion leads to developmental defects**. Next, we were interested to address whether misregulation of the autosomal target genes results in any detectable phenotype in vivo. We noticed that the wings of surviving adult females expressing MSL2tg (*tub-Gal4*, 25 °C) displayed a prominent notch, which was not present in males (Fig. 4a). We therefore tested whether certain cell types and areas of the wing disc are more or less sensitive to the modulation of *msl-2* levels and screened several Gal4 drivers (Fig. 4a, Supplementary Fig. 4c, d). We found female-specific wing defects upon ectopic expression of *msl-2* using *ap-Gal4* and *hh-Gal4*, the latter of which again did not result in the induction of a territory in the wing disc (Supplementary Fig. 4e). These cell-type-specific drivers also allowed us to assess adult phenotypes upon *msl-2* depletion in males (Fig. 4b, c). Upon *msl-2*RNAi, we detected male-specific phenotypes in eyes (*so-Gal4*) and wings (*hh-Gal4*, *vg-Gal4*, *wg-Gal4*, and *ap-Gal4*). *msl-1*RNAi similarly caused a wing phenotype with *hh-Gal4*, while this resulted in male-specific

lethality upon *msl-3*RNAi and *mof*RNAi (Fig. 4d). Consistent with the absence of MLE peaks in ChIP (Fig. 2a), we did not observe any phenotype in *hh-Gal4/mle*RNAi males or females, although this RNAi line was specific and strong enough to cause male-specific lethality with *tub-Gal4* (Fig. 4d, e).

We wondered whether using RNAi would allow us to at least partially uncouple X chromosome-wide from gene-by-gene regulation at autosomal sites. We therefore performed expression analyses in male wing discs using *hh-Gal4* in two different *UAS-msl-2*RNAi lines. We detected *msl-2*, *roX1*, *roX2* and *Klp3a* downregulation, whilst other X-linked (*Ucp4a*, *CG5254*, *socs16D*, and *bnb*) and autosomal control genes were not substantially changed (Fig. 4f and Supplementary Fig. 4f). Remarkably, the autosomal targets *vg*, *ap*, *so*, and *wg* displayed a pronounced and significant downregulation. Concordantly, we detected an upregulation of *vg* and *ap* by ectopic expression of *UAS-msl-2::3Flag* with *hh-Gal4* in females (Fig. 4g). IF stainings performed on male *hh-Gal4/msl-2*RNAi wing discs revealed that the H4K16ac-positive territory was frequently but not always lost in the *hh*-positive cells (Fig. 4h and Supplementary Fig. 4g). Hence, the consistent appearance of the wing phenotype was more directly correlated with expression alterations of autosomal rather than X-linked MSL2 targets. These qPCR experiments do certainly not exclude that some X-linked genes that are not covered in our analyses have important roles in the complex mechanisms operating during morphogenesis. Yet, together with the ChIP data they support the hypothesis that the wing phenotypes are not simply caused by a collective misregulation of the X but are rather an effect of acute alterations on genes directly bound by MSL2.

We were next interested to address whether we could link these phenotypes to aberrant patterning at the cellular level and performed IF on female *hh-Gal4/UAS-msl-2::3Flag* wing discs. Overall, the shape of the disc appeared normal with the typical localization of Wingless (Wg) at the Dorsal-Ventral (DV)

boundary, and MSL2 staining in the posterior, *hh*-expressing compartment (Fig. 4i). However, precisely where the DV boundary encounters the MSL2/*hh*-positive cells, closer inspection showed an interruption of the DV boundary, providing a possible explanation for the strong wing phenotype observed in

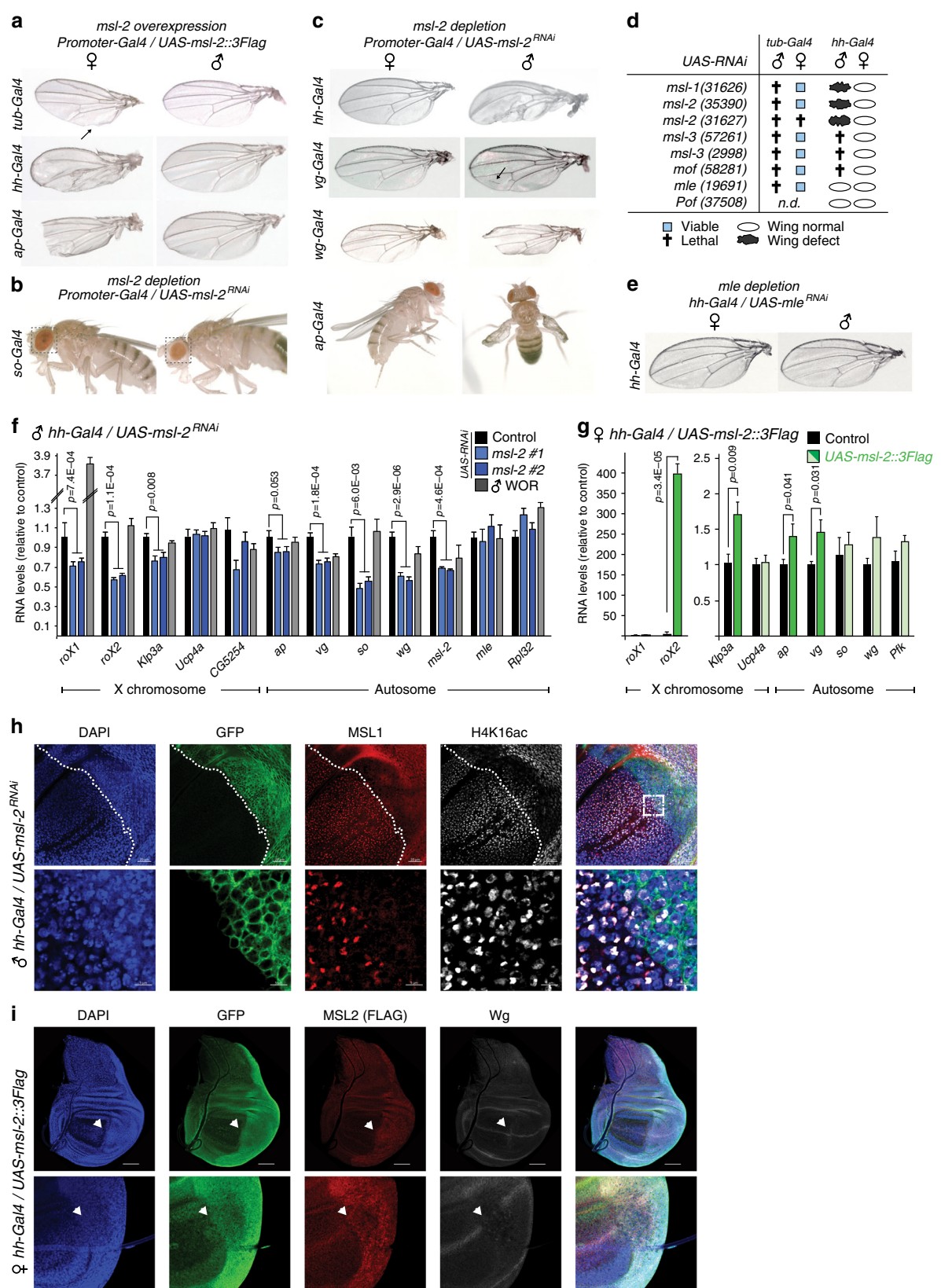

adults. Collectively, our data suggest that MSL2 not only regulates X-chromosomal genes but is also responsible for fine-tuning expression levels of autosomal genes such as *ap* and *wg*. Appropriate regulation of these genes by MSL2 is required for proper development in *Drosophila*.

**Conservation of MSL2-mediated dosage compensation in mammals.** Morphogens involved in embryonic patterning, for example, within the hedgehog or wingless-pathways, are highly conserved from *Drosophila* to higher eukaryotes[35,36]. One could therefore expect common regulatory mechanisms and dosage sensitivity in both systems. The autosomal MSL2 binding in *Drosophila* was indeed reminiscent of MSL2 ChIP in mammalian cells[18,37], in particular because Dhx9, the orthologue of MLE, does not associate with the MSL complex[20]. We therefore created *Msl2* knockout mouse embryonic stem cells (mESCs) allowing us to work in highly standardized conditions, that are required to score subtle regulation (Supplementary Fig. 5a). We obtained two independent lines (*Msl2Δ-A2*, *Msl2Δ-D12*) and confirmed the absence of *Msl2* at the RNA and protein level (Supplementary Fig. 5b, e). Reminiscent to *Drosophila*, where lethality manifests relatively late during development, MSL2 was non-essential for the general maintenance of cellular homeostasis and its absence caused only mild proliferation defects (Supplementary Fig. 5c). Contrary to flies though (Supplementary Fig. 5d), we did not score large changes in bulk H4K16ac levels by IF and western blot and also other histone modifications remained largely unchanged (Supplementary Fig. 5e, f). Given the increase of bulk H4K16ac levels during differentiation[38], we therefore envisioned that H4K16ac might be more locally affected in mESCs and generated H4K16ac ChIP-seq profiles in WT and *Msl2Δ* cells. H4K16ac appeared enriched in large domains encompassing several dozens of active genes and typically correlated with marks present at active gene bodies, for example, H3K36me3 (Supplementary Fig. 5g). In agreement with our hypothesis, H4K16ac in *Msl2Δ* cells was selectively lost from regions surrounding MSL2 peaks, for example, at the *Zfp185* locus, but remained globally unchanged (Supplementary Fig. 5g).

Next, we investigated, how gene expression is affected in *Msl2Δ* mESCs and performed RNA-seq. We were particularly interested in the set of developmental regulatory genes bound on fly autosomes, as we found peaks in the vicinity of the mouse orthologues in our mESC ChIP profiles[18], for example, at *Six1-4* (*Drosophila so*), *Sall1/2* (*Drosophila salm*), or *Tbx20* (*Drosophila mid*) (Fig. 5a). We performed differential gene expression analysis in the two different growth conditions for which we had generated datasets: (1) 2i representing a naive status in a homogeneously tuned environment for mESCs[39] and (2) Serum + LIF representing a primed status characterized by the mosaic expression of pluripotency and differentiation factors

(Supplementary Data 3 and 4). Indeed, the aforementioned genes were significantly downregulated in *Msl2Δ* cells in serum (Fig. 5a). Consistent with the idea that MSL2-mediated H4K16ac impacts on developmental and differentiation, rather than cellular homeostasis pathways, we scored only a small number of differentially expressed (DE) genes upon *Msl2* knockout in 2i medium. Nonetheless, the list of the 16 significantly downregulated genes included not only *Msl2*, but also *Tsix*, for which we have previously shown a regulatory role of the MSLc (Supplementary Data 4)[18]. In contrast to 2i, we scored many more DE genes in mESCs growing in serum-containing medium, where differentiation factors are expressed. DE genes were found on all chromosomes (Supplementary Fig. 5h) and were typically affected by twofold or less (Fig. 5b, c, Supplementary Fig. 5i). Genes, that were downregulated in *Msl2Δ* cells showed decreased H4K16ac levels in ChIP-seq and ChIP-qPCR (Fig. 5d, e), whereas for the upregulated genes the opposite was the case. Finally, the group of genes containing MSL2 peaks at the TSS[18] were downregulated (Fig. 5f).

Next, we performed network analyses on the DE genes, which revealed that they function in the regulation of morphogenesis or pattern specification processes (Supplementary Fig. 5j). Given that *Drosophila* autosomal targets were involved in similar processes, we asked whether the group of 1:1 orthologues bound by MSL2tg in flies is collectively misregulated in mouse ES cells (Fig. 5g). Strikingly, the set of autosomal genes bound in *Drosophila* Cluster 1 (HAS-like motif) was significantly downregulated in *Msl2Δ* mESCs. A random control group containing the same number of genes, orthologous genes from *Drosophila* Cluster 2, 3 (TG-rich motif) or X-linked HAS orthologues were not misregulated (Fig. 5g). Given the small number of DE genes in *Msl2Δ* mESCs, this is intriguing and implies, that Cluster 1 represents an evolutionary ancient set of dosage-sensitive genes, which is regulated by MSL2 in both flies and mice.

Taken together, our data suggests that the final outcome of MSL-mediated regulation in *Drosophila* and mammals is remarkably similar. MSL2-mediated H4K16ac causes a approximately twofold upregulation in gene expression suggesting that as such, the DC function exerted by the MSLc is conserved in both systems. Moreover, DC seems to operate on an ancient set of developmentally regulated genes residing on autosomes, which depend on regulation by MSL2 in both *Drosophila* and mammals.

## Discussion

In this study, we have characterized the outcome of MSL-mediated DC in *Drosophila* and mESCs. We find that the repertoire of dosage-compensated genes in *Drosophila* is much more dynamic in vivo than anticipated and also operates on autosomal genes. At least a subset of these autosomal sites

**Fig. 4** MSL2 depletion leads to developmental defects. **a** Pictures of wings of female and male flies expressing *UAS-msl-2::3Flag* using the indicated Gal4 drivers (see methods, *ap*-Gal4 from G. Pyrowolakis). **b** Pictures of adult female and male flies expressing *UAS-msl-2*^RNAi^ (*BD31627*) with *so*-Gal4. **c** Pictures of wings of female and male flies expressing *UAS-msl-2*^RNAi^ (*BD31627*) with the indicated Gal4 drivers. **d** Scheme displaying the result of viability and wing phenotypes upon *UAS*^RNAi^ of different dosage compensation factors. **e** Pictures of wings of female and male adult flies expressing *UAS-mle*^RNAi^ using *hh*-Gal4. **f** Real-time RT-qPCR analyses of the indicated genes in male wing discs upon *UAS-msl-2*^RNAi^ with *hh*-Gal4. The RNA level of each gene was calculated relative to the geometric mean of *RpL32*, *Pfk* and *U6* expression level. *RNAi* lines for *msl-2* were BD31627 (*msl-2*^RNAi^#1) and BD35390 (*msl-2*^RNAi^#2), WOR refers to wild-type Oregon R. The barplot represents the average of 3–4 independently collected samples each consisting of two wing discs with error bars indicating the SEM. *P*-values were calculated using a one-tailed *t*-test. **g** as in **f** in female wing discs expressing *UAS-msl-2::3Flag* with *hh*-Gal4 (*n* = 5). **h** Immunofluorescence of male wing discs (*UAS-gfp/Y;; hh-Gal4/UAS-msl-2*^RNAi^ *BD35390*) with GFP (green), MSL1 (red) and H4K16ac (white). The bottom panel shows a zoom, DAPI is shown in blue. Scale bar = 50 μM. **i** Immunofluorescence of female wing discs (*UAS-gfp;; hh-Gal4/UAS-msl-2::3Flag*) with GFP (green), MSL2 (FLAG, red), and Wingless (Wg, white). The top and bottom panel show two different wing discs. The arrow points towards regions where the Wg-positive, DV boundary is interrupted. DAPI is shown in blue. Scale bar = 50 μM

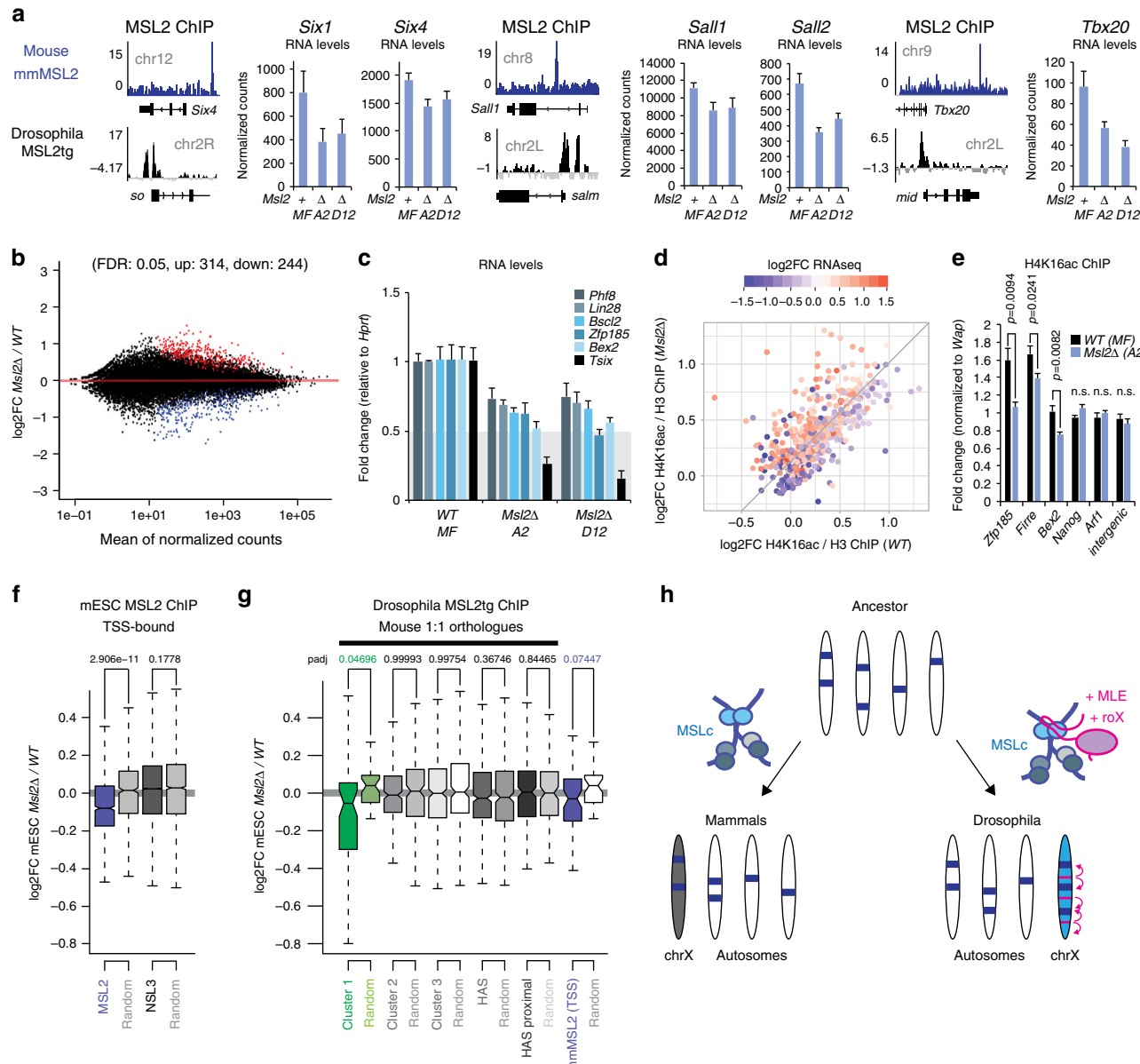

**Fig. 5** Conservation of MSL2-mediated dosage compensation in mammals. **a** Genome-browser snapshot of mESC MSL2 ChIP-seq[18] (top, blue) and *Drosophila* MSL2tg ChIP-seq (bottom, black) at orthologous genes. Barplots show average expression using the normalized read counts by RNA-seq of four biological replicates with error bars indicating SEM. **b** MA plot comparing the mean of the normalized counts versus the log2FC of *Msl2*Δ versus parental mESCs (significantly downregulated and upregulated genes colored in blue and red, respectively (FDR cutoff 0.05)). **c** Real-time RT-qPCR analyses with barplots showing average expression levels relative to *Hprt* of four biological replicates with error bars indicating the SEM. **d** H4K16ac levels on all 558 DE genes identified in RNA-seq. H4K16ac/H3 ChIP enrichment scores were calculated for each DE gene from TSS to TES. The color of the datapoints indicates the log2FC in RNA-seq. Genes below the diagonal lose H4K16ac in *Msl2*Δ, whereas above the diagonal gain *H4K16ac*. **e** ChIP-qPCR analyses in parental and *Msl2*Δ mESCs. The barplot shows the average of four independent biological replicates with error bars indicating the SEM. Enrichment was calculated relative to input. The data are expressed as fold change over the negative control *Wap*. *P*-values were calculated using a one-tailed *t*-test (parental (*WT MF*) versus *Msl2*Δ). **f** Boxplot displaying the log2FC of genes containing a MSL2 peak (1148 genes) or NSL3 peak (5068 genes) within 1 Kb of a TSS versus a random gene set of the same size. *P*-values were calculated using a Welch two sample *t*-test. **g** Boxplot showing the log2FC of the 1:1 mouse orthologues of the *Drosophila* MSL2tg-bound genes in Cluster 1 (66 orthologues), 2 (269 orthologues), 3 (346 orthologues), HAS (432 orthologues), HAS proximal (231 orthologues), or 66 random genes containing mESC TSS peaks. HAS group: genes directly overlapping, HAS proximal group: 266 closest *Drosophila* genes to any HAS (maximum distance of 2344 bp). The log2FC of the respective clusters were tested against 10,000 randomly drawn samples of same size with bootstrapping. *P*-values were corrected for multiple testing by applying a Benjamini–Hochberg correction. **h** Graphical model of the evolution of MSL2-mediated dosage compensation in *Drosophila*

displays a HAS-like motif and is enriched for CLAMP. Therefore, our data may provide an answer to the puzzling observation that the classical, GAGA-rich HAS motif and CLAMP can also be found on autosomes[27,40]. We imagine that MSL binding at GAGA-motif containing genes on autosomes may be cell-type-specific. For example, *ap* is not expressed in the salivary gland, and for this reason may also not require binding and regulation by the MSLc in this context. This is reminiscent of mammalian cells, where MSL2 binding and regulation is dynamically regulated in different cellular contexts[18]. It is important to note that our *Drosophila* profiles represent a sum of all the tissues present in the anterior part of L3 animals. In light of the regulation of developmental genes such as *wg* or *vg*, we envision that it will be important to explore the tissue-specificity of MSL binding and accessibility of its binding sites in greater detail in the future.

In contrast to mammals, MSL2 is sex-specifically expressed in male flies. As MSL2-mediated regulation occurs on genes that have overall similar expression level in both genders this may appear puzzling. However, we discovered that the X-linked *Notch* and *Beadex* have different expression levels in males and females and hence, classify as escape genes. Consequently, shifting the complex regulatory network operating during morphogenesis from a male to female state and vice versa results in developmental defects, for example, in the wing. Moreover, sexual dimorphism is indeed highly prevalent in nature. In flies, this not only includes gametes or abdominal pigmentation, but also visually less obvious examples, for example, the intestinal epithelium or wings[41–43]. It is therefore possible that MSL2 fine-tunes such processes in males. An alternative scenario is, that MSL2 is expressed under certain conditions in females[44,45]. However, our IF and qPCR analyses do not provide any evidence for this hypothesis, implying that if it exists, it is presumably very rare and cell-type specific.

Motifs involved in MSL targeting diverged in Drosophilids and are different in mammals[18,24]. Moreover, MSL2 binding sites in mammals are in some instances located very far away from the genes that are subjected to its regulation classifying them as enhancer-like sites. Yet, our data shows that the ultimate outcome of MSL targeting, acetylation of H4K16 resulting in approximately twofold upregulation appears remarkably similar. Such regulation probably originated from the conserved set of dosage-sensitive genes on autosomes that we have identified. During evolution of sex chromosomes in Dipterans, this elegant system may then have been co-opted for DC of the entire X[46]. In this process, the DC system must have acquired the ability to spread and collectively regulate all X-linked genes, independently of whether they have a direct binding site or not. This feature most likely arises from the association with MLE[15,28], as the absence of MLE binding distinguishes X-linked from evolutionarily conserved sites on autosomes. Moreover, this also distinguishes the mammalian from the *Drosophila* complex[20]. It will be interesting to find out why MLE is recruited to X-linked, but not autosomal sites in *Drosophila*. It is possible that because roX levels accumulate at significantly lower levels at the latter sites, this may not be sufficient for MLE association. Such a scenario is consistent with the fact that although MLE is expressed in females, it does not associate with chromatin in our ChIP data.

Lastly, our study shows that the precise regulation of MSL2 target genes on autosomes is critical in vivo. It has been previously shown that upon chromosomal instability, the MSLc buffers deleterious effects arising from aneuploidy[47]. We expand this picture and provide evidence that the fine-tuned regulation of developmental genes by the MSLc is required for proper patterning and growth, as exemplified by the fly wing. Such regulation also matters in mammals, e.g., for the MSL2 targets *Six1-4* or *Sall1*, whose haploinsufficiency causes syndromes associated with hearing loss, chronic renal failure, limb, and ear anomalies in humans[48–53]. Given the conserved properties of the MSLc as a fine-tuner of gene expression, we believe that interventions aiming at selectively modulating H4K16ac levels might be an interesting perspective in future studies involving dominant mutations in developmental disorders[54].

## Methods

**Drosophila ChIP and FLASH**. One ChIP or FLASH sample typically consisted of 25 wandering L3 larvae, which were crudely dissected with forceps to further process only the front part (carcasses). For ChIP, the fronts were fixed with 1% formaldehyde for 15 min at RT, before stopping the fixation with 0.25 mM Glycine for 5 min at RT. For FLASH, samples were processed directly without formaldehyde fixation. All the following steps were performed at 4 °C with all buffers containing protease inhibitors. Samples were homogenized in extraction buffer G1 (15 mM HEPES (pH 7.4), 20 mM KCl, 10 mM MgCl₂, 0.5% Tween-20, 20% Glycerol). The G1-homogenate was UV crosslinked in a tissue culture plate (0.15 mJ cm⁻² using Vilber Lourmat Bio-Link BLX UV Crosslinker). The extracts were clarified on a sucrose cushion (G1 + 30% sucrose) before washing the nuclei once in NP-40 wash buffer (10 mM Tris (pH 8.0), 5 mM MgCl₂, 1 mM CaCl₂, 10 mM NaCl, 0.1% IGEPAL CA-630, 0.25 M Sucrose). The nuclei were then resuspended in Aline-buffer (20 mM Tris (pH 8.0), 5 mM MgCl₂, 1 mM CaCl₂, 10 mM NaCl, 1% Triton-X100, 0.25 M Sucrose). For ChIP, MNase digestion was performed at 25 °C for 15 min typically using 0.5–1 µL Micrococcal Nuclease (M0247, New England Biolabs NEB, amounts optimized depending on the batch). The MNase digestion step was skipped for FLASH. 33 µL of 10×High Salt Buffer was added (200 mM EDTA, 4 M NaCl) before treatment in a Bioruptor Pico (10 cycles, 30 s ON/OFF). The extracts were clarified by centrifugation for 5 min at 12,000 × g. These extracts served as starting material for ChIP or FLASH. For ChIP, samples were immunoprecipitated according to standard procedures (refer to Supplementary Data 1). After reverse crosslinking in 1×TE at 65 °C for 16 h, RNaseA treatment and Proteinase K-treatment, the DNA was phenol–chloroform extracted followed by ethanol precipitation. This material was used for qPCR analyses or deep sequencing. Library preparation for sequencing was performed using a NEBNext® Ultra™ II DNA Library Prep Kit for Illumina (E7645, New England Biolabs NEB) according to the recommendations by Skene and Henik-off[22]. For FLASH, the IPs were processed according to[20]. Briefly, the washed and IPed material was treated with RNaseI before 3′-end repair with PNK and ligation of FLASH adapters with T4 RNA Ligase 1 (1 h at 25 °C). The RNA was eluted from the beads using Proteinase K treatment and purification using an Oligo Clean and Concentrator Kit (Zymo Research). Isolated RNA was reverse-transcribed using SuperScript3 (Thermo Fischer) at 42, 50, and 55 °C for 20 min each, followed by RNase H treatment. The cDNA was column-purified using Olico Clean and Concentrator and circularized with CircLigase (Epicentre) for 16 h. Circularized cDNA was directly PCR amplified and sequenced on Illumina NextSeq 500 in paired-end mode.

**Mouse ES cell ChIP**. Cells grown in 15 cm dishes were fixed in PBS containing 1% methanol-free formaldehyde (Thermo scientific #28906) for 8 min at RT. Chromatin was purified using Paro-washes as described in ref. [55] Chromatin was fragmented using MNase as for the *Drosophila* ChIP protocol but digesting at 37 °C and adding SDS to 0.1% final concentration after addition of EDTA-containing buffer. Cleared chromatin was incubated for 6–8 h with antibody, before addition of Protein A dynabeads for 1 h. IPs and chromatin were further processed as for *Drosophila* ChIP-seq.

**Bioinformatic resources**. bbmerge (https://jgi.doe.gov/data-and-tools/bbtools/)
Bowtie2[56] (https://github.com/BenLangmead/bowtie2)
MACS2[57] (https://github.com/taoliu/MACS/tree/master/MACS2)
deepTools2[58] (https://deeptools.readthedocs.io/en/latest/)
BEDTools[59] (http://bedtools.readthedocs.io/en/latest/)
SAMTools[60] (http://samtools.sourceforge.net)
liftOver (https://genome.ucsc.edu/cgi-bin/hgLiftOver)
Galaxy[61] (https://github.com/bgruening/galaxytools)
IGV[62]
R (https://www.r-project.org)
MEME[63]
DESeq2[64] (http://bioconductor.org/packages/DESeq2/)

**Genomes and annotations**. *D. melanogaster*, dm6 assembly and annotations from (www.flybase.org)
*M. musculus*, GRCm38–mm10 assembly and annotations (www.ensembl.org)
*D. melanogaster* High-Affinity Sites (HAS) peaks definition was taken from GSE69208 based on roX ChIRP
Orthologous gene lists were generated on Ensembl biomart.

**ChIP-seq datasets and antibodies**. An overview of all ChIP-seq datasets generated including mapping statistics and antibodies is provided in Supplementary Data 1.

**Processing of ChIP-seq datasets**. ChIP-seq datasets were mapped to dm6 or mm10 in paired-end mode with bowtie2 (Galaxy Version 2.3.0.1) using parameters:--sensitive, --end-to-end for histones and --sensitive, --local for all other profiles. For MSL2tg ChIP-seq, read-pairs with a minimal overlap of 10 bp were merged and Illumina adapters were trimmed using bbmerge (v7.3), resulting in small (≤140 bp) and large (>140 bp) fragments. Merged, single-end reads (≤140 bp) and unmerged, paired-end reads (>140 bp) were then mapped separately using bowtie2. For all other ChIP-seq datasets, paired-end reads were directly mapped. Peaks were called with MACS2 (Galaxy Version 2.1.1.20160309.0) on each replicate and merged using cat, BEDtools sort (Galaxy Version 2.27.0.0) followed by BEDtools merge with 10 bp option. For motif analysis we used MEME (Galaxy Version 4.11.1.0) with default settings. Coverage files were generated with deepTools2 (Galaxy Version 2.5.1.0.0) bamCompare, binsize of 2. Duplicate reads and reads with a mapping quality <10 were removed, the X chromosome was ignored for scaling. The data were normalized as follows: log2FC over Input for *Drosophila* H3, H4K16ac, H3K36me3, H4ac, H3ac; Input subtraction for *Drosophila* MSL2-Flag (MSL2tg), untagged-Flag, MLE, MSL1, NSL1 (S2 cells), MSL2 (S2 cells), CLAMP (larvae); 1 × Coverage for *Drosophila* H4K16ac, H3K36me3, H3ac, H4ac shown in Figs. 1d or 2a, respectively; mESC H4K16ac log2FC over H3. For plotting and illustrative purposes, the BAM-files of biological replicates were merged using SAMTools (Galaxy Version 1.2.0) for further processing of the data. Heatmaps were generated using deepTools2 computeMatrix and plotHeatmap. Enrichment scores were calculated using deepTools2 multiBigwigSummary. Genome-browser snapshots were generated using IGV. Genes associated with MSL2tg peaks at promoters are listed in Supplementary Data 2.

**Processing of FLASH datasets**. Forward and reverse reads having a minimum overlap of 30 nt were merged and Illumina adapters were trimmed using bbmerge (v.7.3). Libraries were demultiplexed using the barcode contained in the custommade adapters allowing uniquely identifiable barcodes and a maximum hamming distance of 2. Merged reads and forward unmerged reads were mapped in single-end mode to the reference genome (dm6) using bowtie2 (v2.2.3) with parameters:--sensitive –end-to-end. Uniquely mapped reads were identified by having a mapping quality >10 and were subjected to PCR-duplicate removal based on the random tags contained in the custom-made adapters using a custom-made Python script. Coverage files of uniquely mapped, PCR duplicates removed reads were generated using deepTools (v.2.4.1).

**Processing of RNA-seq datasets**. Poly(A) RNA-seq data was mapped to the mm10 genome using HISAT2 (v2.2.0-beta) using default parameters. Primary alignments were used for counting (featureCounts, v1.4.6-p2) and differential expression analysis (DESeq2, v1.8.2). Only genes with FDR < 0.05 in both *Msl2* knockout clones (A2 and D12 as replicates) were considered differentially expressed. Differentially expressed genes are listed in Supplementary Data 3 and 4.

**Other data analyses**. Boxplots were generated in R with default settings. The line that divides the box into two parts represents the median, the ends the upper and lower quartiles (first and third quartile). For GO Term analyses in *Drosophila*, we used the PANTHER Overrepresentation Test (release 20170413, Bonferroni correction = TRUE) on www.geneontology.org/.

Pathway analysis for mESCs was performed using GSEA Release 6.1. Preranked list of DE genes by log2FC ($p_{adj}$ < 0.05) was run with GSEA using standard weighted statistical analysis on cp gene sets. Enrichment map visualization was obtained using Cytoscape (release 3.5.0). Barplots are presented as average ± standard error of the mean (SEM), the individual datapoints are provided in Supplementary Data 5.

**Cloning and gRNAs**. gRNAs targeting *Drosophila* msl-2 (5′-GCATGTGTAACT-GAGCTCCTA and 3′-gCGAGGAGATCATGTCGGGCT, g = non-templated for efficient U6-driven transcription) were cloned by annealing two phosphorylated complementary oligos into the BbsI site of pBFv-U6.2B (flybase FBmc0003128). For C-terminal tagging of *msl-2*, we used the 3′ gRNA and a donor plasmid for homologous recombination in a pJET1.2 backbone. The donor encompasses the *msl-2* CDS followed by a 3xHA-TEVc-Bio-6xHis tag sequence and 2.5 kb of the 3′ UTR (TEVc: TEV cleavage site, Bio-tag[65]), which was amplified from cDNA.

gRNAs targeting mouse *Msl2* (5′gACGTTTCTCTTCCGACGGCG and g-ttaggcggacttcgaactag) (g = non-templated for efficient U6-driven transcription) were cloned by annealing two phosphorylated complementary oligos into the *Bbs*I site of a PX-459 derivative (pSpCas9(BB)-2A-Puro, Addgene 62988), where the Puro-selection cassette was removed (EcoRI) and the Flag-tag was replaced with an HA-tag. We used a split Puromycin resistance approach for selection[66].

The molecular nature of all CRISPR/Cas9-generated alleles in this study was verified by PCR (outside homology arms) and pJET cloning/Sanger sequencing.

**Cell culture**. All cell culture was performed in a humidified incubator at 37 °C and 5% $CO_2$. The parental male mouse embryonic stem cells (WT MF) were kindly donated by the Bühler laboratory, FMI Basel[66], and cultivated on Attachment Factor in two different ESC culture media.

2i media: KnockOutTM DMEM supplemented with 15% KnockOutTM Serum Replacement, 0.1 mM MEAA, 1 mM Na Pyruvate, 0.1 mM 2-mercaptoethanol, 4 mM GlutaMAXTM, 5 μg mL$^{-1}$ Insulin (Sigma I0516), 50 U mL$^{-1}$ Pen-Strep, 200 U mL$^{-1}$ LIF, 1 μM PD0325901 (StemGent 04-0006), 3 μM CHIR99021 (StemGent 04-0003)

Serum media: DMEM supplemented with 15% FCS (PAN, PANSera Lot n. P1302077ES), 0.1 mM MEAA, 1 nM Na Pyruvate, 0.1 mM 2-mercaptoethanol, 4 mM GlutaMAXTM, 50 U mL$^{-1}$ Pen/Strep, 400 U mL$^{-1}$ LIF (ESGRO #ESG1106).

**Antibodies**. ChIP-seq and FLASH: Antibodies and concentrations are provided in Supplementary Data 1.

For western blots (Drosophila): anti-MSL1 (Rabbit, validated in ref. [9], 1:3000), anti-MSL2 (Rabbit, Santa Cruz sc-32459, 1:1000), anti-MOF (Rabbit, validated in ref. [9], 1:2000), anti-MSL3 (Rat, validated in ref. [9], 1:1000). For western blots (mESCs): anti-Msl2 (Rabbit, HPA003413 Sigma, 1:1000), anti-Mof (Rabbit, A3000992A BETHYL Montgomery, TX, 1:1000). For western blots (General): anti-H4K16ac (Rabbit, Milipore 07-329, 1:3000), anti-H3 (Mouse, Active Motif 39763 1:5000), anti-H4ac (pan) (Rabbit, Milipore 06-598, 1:1000), anti-H4K12ac (Rabbit, Milipore 07-595, 1:1000), anti-H4K5ac (Rabbit, Milipore, 07-327, 1:1000), anti-H3K9me2 (Rabbit, Active Motif, 39239, 1:1000), anti-H4K20me1 (Rabbit, Abcam ab9051, 1:1000), anti-H4 (Active Motif 61521, 1:1000), anti-HA (Mouse, Covance #MMS-101P, 1:5000), anti-RPB3 (Rabbit, 1:2000, generated in Akhtar lab), anti-Tubulin (Mouse, abcam 44928, 1:5000), FLAG-HRP (Sigma, SAB4200119 Sigma, 1:10,000), HA-HRP (Sigma, 12013819001 Roche, 1:5000), anti-RNA Pol2 (Mouse clone 4H8, 101307, Active Motif, 1:10,000). Secondary antibodies (1:10,000) used were anti-mouse IgG HRP (NXA931), anti-rat IgG HRP (NA935V), and anti-rabbit IgG HRP (NA934) from Sigma. For immunofluorescence (IF): anti-MSL3 (Rat, validated in ref. [9], 1:200), anti-HA (Mouse, Covance #MMS-101P, 1:400 for squashes, 1:150 for wing discs), anti-MOF (Rabbit, validated in ref. [9], 1:300), anti-MSL1 (Rabbit, validated in ref. [9], 1:300; Rat, validated in ref. [10], 1:300), anti-FLAG (Rat, Clone L5, Biolegend 637301, 1:200 for squashes, 1:250 for wing discs), anti-Wg (DSHB Hybridoma Bank 4D4, 1:150), anti-H4K16ac (Rabbit, Active Motif 39167, 1:200 for Fig. 4i, Supplementary Fig. 4e), anti-H4K16ac (Rabbit, Milipore 07-329, 1:500 for mESC IF), anti-H4K16ac (Mouse Monoclonal, Active Motif, # 61529 1:300 for Fig. 4h, Supplementary Fig. 4g), anti-GFP (Rabbit, Thermo Fisher PA1-980A, 1:200 for wing discs), anti-E-cadherin (BD Bioscience, 610181 1:200), anti-RNA pol II (Mouse clone 4H8, 101307, Active Motif, 1:1000 for squashes), Alexa secondary antibodies were purchased from Thermo Fisher and used at 1:500.

**Preparation of protein extracts**. Ten adult male or female flies aged 12–24 h after eclosion were collected and placed on ice. Flies were decapitated, and heads were homogenized in 50 μL 1 × Roti-Load reducing sample buffer (Roth). Homogenates were incubated for 5 min at 70 °C, and typically 10–20 μL homogenate used for SDS–PAGE. Nuclear extracts from mESCs were prepared using a NE-PER kit (Thermo Fisher, 78833).

**Western blot**. Proteins were separated by regular SDS–PAGE, transferred overnight at 60 mA to PVDF membranes using a Bio-Rad Wet Tank Blotting System in Towbin-Buffer containing 10% methanol. The membrane was blocked for 30 min in 5% milk in TBS-0.1%Tween before incubation with antibodies in 0.5% milk TBS-0.1% Tween (typically 5–6 h at RT). Secondary HRP-coupled antibodies were used at 1:10,000 (1 h). Blots were developed using Lumi-Light Western Blotting substrate (Sigma, 12015200001 ROCHE) and imaged on a ChemiDoc XRS + (Bio-Rad). Uncropped blots are displayed in Supplementary Fig. 6.

**RNA expression analysis**. Fly tissues and mouse ES cells were homogenized in TRIzol™ Reagent (Thermo Fisher 15596026) followed by RNA purification using a Direct-zol™ RNA MiniPrep kit (Zymo Research, R2050) according to the manufacturers instruction. cDNA was synthesized from 1000 ng (whole larvae, mES or S2 cells) or 150–200 ng (wing discs, larval brains) of total RNA isolated from each tissue. The GoScript™ Reverse Transcription System with Random Primers was used for cDNA synthesis according to the manufacturers instruction. For stranded polyA + mRNA-Seq Library Preparation, TruSeq stranded mRNA sample preparation kit (RS-122-2101, Illumina) was used.

**Quantitative real-time PCR**. qPCR was performed on a Roche LightCycler® II using FastStart Universal SYBR Green Master (Roche, 04913914001) in a 7 μL reaction at 300 nM final concentration of each primer. Cycling conditions as recommended by the manufacturer were used. We corrected for primer efficiency using serial dilutions. Experiments were conducted using at least three independently collected biological replicates. Primers sequences are listed in Supplementary Table 1.

**Drosophila rearing conditions**. *Drosophila melanogaster* were reared on a cornflour-molasses fruit fly medium [1 L water, 12 g agar–agar threads, 18 g bakery yeast, 10 g soya flour, 80 g cornflour, 22 g molasses, 80 g malt extract, 2.4 g 4-hydroxibenzoic acid methylester (Nipagin), 6.25 mL propionic acid] at 25 °C, 70% relative humidity and 12 h dark/12 h light cycle. All experimental crosses and experiments were conducted at 25 °C.

**Drosophila stocks**. The following stocks were obtained from the Bloomington *Drosophila* Stock Center or were kindly donated:

$y^1$ $w^*$; P{tubP-GAL4}LL7/TM3, $Sb^1$ (BDSC #5138),
msl-2$^{kmA}$/CyO (BDSC #25158)
msl-2$^{227}$, $bw^1$/CyO (BDSC #5871)
mle$^9$, $cn^1$, $bw^1$/CyO (BDSC #5873)
$y^*$, $w^*$; hs-FLP, UAS-GFP;; hh-Gal4/TM6BTb (Georgios Pyrowolakis, University of Freiburg, Germany)
$y^*$, $w^*$; ap-Gal4/CyO, (Georgios Pyrowolakis, University of Freiburg, Germany)
$y^*$, $w^*$;; C765-Gal4/TM3, $Sb^1$, $Ser^*$ (Georgios Pyrowolakis, University of Freiburg, Germany)
$w^*$; nub-Gal4, UAS-GFP (Georgios Pyrowolakis, University of Freiburg, Germany)
$y^*$, $w^*$;; ci-Gal4 (III) (Georgios Pyrowolakis, University of Freiburg, Germany)
$w^*$; ptc-Gal4 (II) (BDSC #2017)
en-gal4 (II) (Georgios Pyrowolakis, University of Freiburg, Germany)
A9-Gal4 (III) (Georgios Pyrowolakis, University of Freiburg, Germany)
$y^1$, $w^*$; P{w[+mC]=Act5C-Gal4}25FO1/CyO, $y^+$ (BDSC #4414)
$w^*$; P{2[+$m^*$]=Ubi-Gal4.U}2/CyO (BDSC #32551)
$y^1$, $w^{1118}$; P{w[+mC]-vgMQ-Gal4.Exel}2 (BDSC #8230)
$y^1$, $w^{1118}$; P{w[+mW.hs]=GawB}ap[md544]/CyO (BDSC #3041)
$y^1$, $v^1$;; P{nos-Cas9.TH00787.N}attP2 (Fillip Port, LMB, Cambridge, UK)
$y^1$, M{Act5C-Cas9.P.RFP}ZH-2A $w^{1118}$ Lig4$^{169}$ (BDSC #58492)
$y^1$, $w^*$, wg-Gal4, UAS-GFP/CyO (Sergio Casas Tinto, Cajal Institute, Madrid, Spain)
$y^1$, $w^*$; P{so7-GAL4}A (BDSC #26810)
$y^1$, $v^1$; P{TRiP.JF01411}attP2 (BDSC #31626, UAS-msl-1$^{RNAi}$)
$y^1$, $v^1$; P{TRiP.JF01412}attP2 (BDSC #31627, UAS-msl-2$^{RNAi}$)
$y^1$, $sc^*$, $v^1$; P{TRiP.GL00309}attP2/TM3, $Sb^1$ (BDSC #35390, UAS-msl-2$^{RNAi}$)
$y^1$, $sc^*$, $v^1$; P{TRiP.HMC04654}attP40 (BDSC #57261, UAS-msl-3$^{RNAi}$)
$w^{1118}$; P{GD1492}v2998 (VDRC #2998, UAS-msl-3$^{RNAi}$)
$w^{1118}$; P{GD1488}v19691 (VDRC #19691, UAS-mle$^{RNAi}$)
$y^1$, $sc^*$, $v^1$; P{TRiP.HMS01650}attP40 (BDSC #37508, UAS-Pof$^{RNAi}$)
$y^1$, $v^1$; P{TRiP.HMJ22366}attP40/CyO (BDSC #58281, UAS-mof$^{RNAi}$)
White-eyed ($w^{1118}$) Oregon R was used as a wild-type *Drosophila melanogaster* strain.
All lines used in this study were generated via standard genetic crosses from the above listed stocks.

**Integrase-mediated generation of UAS-msl-2::3Flag**. The transgenic lines carrying C-terminally 3×FLAG-tagged wild-type *msl-2* under the control of a UAS promoter were generated through phiC31 integrase-mediated germline transformation as previously described[67]. Plasmid DNA was injected into *y1 M{vas-int. Dm}ZH-2A w*; PBac{y+-attP-3B}VK00033* embryos (BDSC #24871), that carry an attP docking site at position 65B2 on chromosome arm 3 L[68] and a *Drosophila* codon-optimized ΦC31 integrase driven in the germline by the vasa promoter[69].

**CRISPR/Cas9-generated *msl-2* knockout (*msl-2$^\Delta$* alleles)**. To generate an *msl-2* loss-of-function allele that lacks the complete ORF, constructs expressing gRNAs (see cloning) were first injected in 20 *Act5c-Cas9* embryos to test for their efficiency. Twenty-four hours after injection five embryos were used for single embryo gDNA preparation as previously described[70]. Screening of defined deletion events was performed by PCR. The couple of gRNAs with the highest efficiency was subsequently used for injection of *nos-Cas9.TH00787.N* embryos. The injected flies were outcrossed individually with *yw; CyO, Act5c-GFP/If* (second chromosome balancer, derivative of BDSC #4533). Ten individual males from the progeny of each cross were backcrossed to three virgin *yw; CyO, Act5c-GFP/If* females and after 4 days used for single fly gDNA preparation and PCR screened for deletion events. Subsequently sequencing confirmed the targeted generation of a precise deletion of the *msl-2* ORF. Heterozygous balanced flies were collected to establish stocks and *msl-2* loss-of-function was further verified by expression analysis and the characteristic male-specific lethality. Two lines, namely *yw; msl-2$^{\Delta 7}$/CyO, Act5C-GFP* and *yw; msl-2$^{\Delta 10}$/CyO, Act5C-GFP* were used in this study.

**CRIPSR/Cas9-mediated generation of *msl-2::3HA* allele**. *nos-Cas9.TH00787.N* embryos were co-injected the gRNA and donor plasmid described in the cloning paragraph above at 75 ng/100 ng, respectively[71]. The progeny of the injected flies was subjected to the same crosses and analysis as above. A targeted event was recovered and as stock was established that was homozygous viable but sterile, presumably due to a secondary off-target mutation. Subsequently, using standard genetic techniques, the sterility-inducing mutation was recombined out and a homozygous viable and fertile stock was established and used in this study.

**Immunofluorescence and microscopy**. Antibody concentrations are described in the antibodies section. Polytene chromosomes from L3 larvae were prepared as described in ref.[72] Imaginal discs were stained according to standard procedures. Briefly, inverted larvae were fixed in 4% formaldehyde/PEM (0.1 M PIPES pH 6.9, 1 mM EGTA, 1 mM MgCl$_2$), washed three times with PBS (0.3% Triton-X100), blocked in 10% goat serum (0.3% Triton-X100) and stained overnight with primary antibody. After secondary antibody incubation, samples were thoroughly washed with PBS (0.3% Triton-X100) before dissection of wing discs and mounting. Slides where mounted used ProLong™ Gold Antifade Mountant with DAPI (Thermo Fisher P36935).

mESCs were plated on gelatin-coated ibidi slides (AF GIBCO #S006100, ibidi #80826). They were washed in ice-cold PBS and fixed in 4% formaldehyde for 15 min on ice, followed by a 0.25% Triton-X100/PBS permeabilization step for 3 min. After blocking in 1% BSA, mESCs were incubated overnight in primary antibody solution. Secondary antibody incubation was performed for 1 h. Mounting was performed using Polyvinyl alcohol mounting medium with DABCO, antifading (#10981 Sigma).

Polytene squashes and mESCs were imaged on a Zeiss Spinning Disk Confocal microscope. Wing discs were imaged on a Zeiss LSM 880 using Airyscan/Super Resolution setting mode. High-resolution pictures were obtained using a 63x objective and tile scanning with 5% overlap of the full wing disc, followed by stitching post-processing in ZEN blue software after Airyscan processing and deconvolution.

**Cell proliferation analysis**. Parental (WT MF) and *Msl2Δ* (A2/D12) cells were seeded in equal numbers at day 0 and grown for 4 days in culture medium. After 4 days, the total number of cells in each well was counted.

## Data availability

The ChIP-seq, RNA-seq, and FLASH data generated in this study have been deposited in the Gene Expression Omnibus database (GSE109901). Additional datasets from other publications are listed in Supplementary Data 1.

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

## Acknowledgements

We wish to thank all labs for kindly sharing fly stocks. We thank Marc Bühler for sharing the parental mESCs used in this study and Tuncay Baubec for advise on ChIP. We acknowledge the help of Mariana Schulte and Gina Renschler in optimizing the ChIP protocol, Christian Kämmerer for the *msl-2* gRNA plasmid, Josip Hermann for *Msl2Δ*

mESC characterizations, Aline Gaub for polytene squashes, and Ibrahim A. Ilik for
FLASH. We would also like to thank Ulrike Boenisch and her team from the MPI core
NGS facility and Devon Ryan, Fidel Ramirez, and Gautier Richard for support with NGS
data analysis. We are grateful to Aline Gaub, Ibrahim A. Ilik, and Raed Hmadi for critical
reading of this manuscript. CIKV was supported by DDV and a Human Frontier Science
Program (HFSP) long-term fellowship (000233/2014-L). This work was supported by
CRC992, CRC1140, and CRC746 awarded to AA.

## Author contributions

C.I.K.V. characterized fly lines, analyzed wing phenotypes, developed the ChIP/FLASH
protocol, generated all ChIP data and libraries, analyzed ChIP-seq data, analyzed RNA-
seq data, performed expression analyses, western blots, and generated all plasmids used
in this study. M.F.B. created, cultured, and characterized *Msl2Δ* mESCs, and performed
all immunofluorescence (IF) and microscopy. G.S. generated FLASH libraries, and
analyzed FLASH and RNA-seq data. N.M.G. and P.G. generated, characterized and
maintained fly lines, and collected starting material for ChIP/FLASH/IF experiments and
expression analyses. P.G. performed all *Drosophila* crosses and genetic experiments. C.I.
K.V. wrote the manuscript with input from all authors. A.A. supervised the study,
provided guidance, and funding.

## Additional information

018-05642-2.

**Competing interests:** The authors declare no competing interests.

**Reprints and permission** information is available online at http://npg.nature.com/
reprintsandpermissions/

**Publisher's note:** Springer Nature remains neutral with regard to jurisdictional claims in
published maps and institutional affiliations.

