## [Peer Review File · Nature Communications]

Reviewers' comments:

Reviewer #1 (Remarks to the Author):

SUMMARY

The current manuscript from Valsecchi et al. presents the novel and intriguing finding that MSL2 (or a MSL complex lacking MLE) in addition to the function on the X-chromosome also targets dosage sensitive autosomal genes involved in patterning and morphogenesis. MSL2 is only expressed in males and changes in amounts of the MSL-complex in males will have dramatic consequences on expression of X-chromosomal genes. This leads to an apparent challenge to distinguish between the direct effects on these autosomal targets and the indirect effects caused by a deregulated X-chromosome. The author claims that the precise regulation of these genes by MSL2 is required for proper development of the fly wing and provide support that MSL2 mediated regulation of this set of genes is conserved in evolution.

The paper is based on a large amount of data and the presented results are of general importance, relevance and interest. In particular it provides an intriguing hypothesis on the origin of the MSL2 mediated dosage compensation system and how this system may have been co-opted for dosage compensation during evolution of sex chromosomes.

CRITIQUE my main concerns are whether the used MSL2tg construct reproduce endogenous MSL2 expression (point 2 below) and whether the statement "that the precise regulation of these genes by MSL2 is required for proper development of the fly wing" is supported by the presented data (point 5 below):

1. **MINOR:** It is stated in the paper, both in the abstract and the discussion, that the MSLc mediates two-fold upregulation of the single male chromosome. Although often claimed this statement, to my knowledge, still lacks experimental support. The male X-chromosome is upregulated two-fold but the MSLc mediates only part of this, as far as been experimentally shown, see e.g. Hamada et al. 2005, Straub et al. 2008, Zhang et al. 2010, Deng et al. 2009 as well as from the authors lab Vaquerizas et al. 2013. Intriguing recent theoretical arguments state that relation between e.g. RNP2 and mRNA expression is non-linear (Dasmeh et al. 2017). This might turn out to be correct but at this point this needs further experimental support.

2. **MAJOR:** The presented line of arguments is highly dependent on the assumption that MSL2tg (UAS-msl2.FLAG driven by tub.GAL4) reproduce the endogenous expression of MSL2 and the endogenous binding pattern. I am very surprised by the low X-specificity presented in Figure 1a and I would therefore suggest much stronger support that MSL2tg (UAS-msl2.FLAG driven by tub.GAL4) is a reliable proxy for endogenous MSL2 (in males). Examples of data to better support this assumption:

a. Polytene chromosome stainings with this transgene. Is it as X-specific as the endogenous MSL2? Please, also include polytene stainings for females expressing MSL2tg since these are viable to adult stage in contrast to previously published msl2 transgenes expressed in females.

b. How comes (if I understand it correct) that expressing MSL2tg in females doesn't cause lethality?

c. As presented, the ChIP profiles are not fully convincing. Please show profiles over larger regions with comparisons of ChIP profiles for MSL2tg with endogenous MSL2 (e.g. from Straub et al.) or endogenous MSL1, MSL3 or MOF. Show profiles for e.g. 1Mb of the X-chromosome and 1Mb from an autosome including some of the autosomal genes of interest. Use the same scale for X and autosomes for each profile.

3. Presenting ChIP profiles (and ChIRP profiles)

a. In Figure 1b, 2f, 4a please use the same scale on the y-axis (for one specific experiment) as much as possible. For example in Figure 1b it is impossible to judge if the roX2 peaks in wg and en are significant. If different scales are required this needs to be clearly pointed out in e.g. the figure legend. Please also ensure that the information on experiments are included in the figure legend. For example, I assume that roX2 in figure 1b refers to the roX2 ChIRP from Quinn et al 2016 but

this may be wrong and it is not indicated in the legend.

b. Concerning the y-axis scale for ChIP and ChIRP profiles – what is actually shown? I understand it shows enrichment and not in a log₂ scale but if so how comes the negative values?

c. To be able to judge the significance of the peaks shown in ChIP and ChIRP profiles please provide figures on zoomed out browser views on e.g. 1Mb from the X-chromosome and from an autosome with the same y-axis scale for comparison.

d. ChIP qPCR results to support are shown in SFig 2d. Please show this as “% of input”. The y-axis scale “fold change over non targets” is not very informative on the ChIP quality and that HAS and “target genes” are enriched just 2-4 fold more than the control amplicons ent2 and CG15011 is not in line with previously published enrichments.

4. Classification of targeted genes: 49 genes are classified as “cluster 1”, 176 as cluster 2 and 283 as “cluster 3” in total 508 genes.

a. The number of targeted genes (508) makes it important to clarify the logics in which “example genes” that are shown. The genes wg, en, ap, so, mirr and opa are all classified as cluster 1 while vg is classified as cluster 3. Please state this in the text and explain why including these specific genes.

5. MAJOR: One important statement in the paper is that the precise regulation of these genes by MSL2 (the autosomal ones like vg, wg, en etc.) is required for proper development of the fly wing. This is a novel important finding but also a statement with some caveats.

a. Figure 3c shows that overexpression of msl2 in females causes a wing defect (but not lethality). Maybe I am wrong in this but to me an induction of MSL2 in females should increase expression levels on many genes (thousands) and it is impossible to distinguish direct effects from the suggested target genes from indirect effects from e.g. X-chromosomal genes.

b. Figure 3e shows wing defects in males when msl2 is depleted through RNAi. Again, msl2 depletion should cause a changed expression levels of many genes and it is not possible to distinguish direct from indirect effect (and X-chromosome targets from autosomal targets). However, to control for this the authors show that depletion of mle will not give the same phenotype. This becomes a critical experiment and in my opinion much more support is needed here. As I understand the paper the prediction would be that depletion of msl2, msl1, msl3 or mof should generate the wing phenotype while depletion of mle should not under the assumption that the RNAi for all these genes is similar in efficiency. My question is how the X-chromosomal effect can be convincingly separated from the “autosomal target effect”?

ADDITIONAL COMMENTS:

6. On page 3 line 5 the abbreviation “DC” is used for the first time but without explanation, please correct.

7. Just out of curiosity – why was a ChIP-seq using the MSL2 antibody on wildtype males not included to assess how well MSL2tg reproduce endogenous expression in males?

8. Page 6 line 11: “Remarkably, the H4K16ac pattern upon deletion of msl-2 in males was looked essentially identical to the one in wild-type females”. Why is this “Remarkable”? Isn’t it what we would expect?

9. Page 6 last sentence: “indicating that the global regulation of the X is presumably invariant in different tissues”. Please explain this statement in relation to the claim that “accessibility of these sites is cell-type specific”.

10. Please be consistent in naming profiles. For example, in Figure 1 the MSL2tg profile is named MSL2tg but in Figure 2f the profile is named MSL2 and in Figure 3 dmMSL2

11. Please review and correct genetic nomenclature. For example in Fig 3f the genotype is stated as hh-GAL4/UAS-GFP, UAS-msl2::Flag3. I don’t understand this. Was the UAS-GFP transgene

recombined into the UAS-*msl2::Flag3* chromosome and if so which UAS-GFP was used and was this setting used in other experiments as well?

12. Sfig1a: MSL3 shows a double band in the control but not in the other genotypes – why?

13. Sfig1c: What are the H4K16ac and roX2 profiles – which experiments?

14. Sfig3d: Why do we see GFP staining in the 3d panel? According to the genotype stated there is no UAS-GFP in this line.

15. Supplementary figure 3e is supposed to show that MSL2^{tg} does not localize to a typical H4K16ac-positive territory characteristic for the male X. The resolution of the provided figure is not enough to draw this conclusion.

Reviewer #2 (Remarks to the Author):

Nature Communications manuscript NCOMMS-18-04434-T

This study starts with the very interesting premise that MSL2 orchestrates developmental gene regulation on the autosomes of flies, and that regulation of developmental genes by MSL2 is evolutionarily conserved in mammals. The normal function of MSL orthologs in mammals is relatively poorly studied, making this an advance. This idea could explain the binding of MSL proteins to a number of well-known autosomal sites of flies. However, many of the conclusions are flawed by potentially qualifying technical issues. Furthermore, aspects of the current findings disagree with prior studies in trivial or significant ways. These differences are not acknowledged or discussed.

The story in mouse cells is interesting and parallels with flies intriguing. While the foundational premise of this work is interesting (and may ultimately be biologically important and exciting) the current manuscript is reaching for a story that is not quite there yet.

Specific criticisms

Autosomal MSL binding sites have been previously studied. No effort is made to determine if sites currently identified correspond to ones mapped previously.

The discovery that ectopic MSL2 expression elicits inappropriate dosage compensation in females is over 20 years old, but is presented as novel (p.5).

The authors maintain that MSL2 binds autosomal chromatin, but MLE does not. This contradicts previous findings that MLE is found throughout the genome of females, and only becomes restricted to the X in males (Genetics 172, 963 and references therein). Furthermore, MLE is tethered to chromatin by RNA. MNase is used for chromatin preparation in this study, selectively releasing MLE.

Bx is indeed dosage compensated. This gene is responsible for characteristic wing notching in triplo-X females, as well as in females that ectopically express MSL2 (Genetics 183, 811). Increased expression of Bx is almost certainly the source of the defects that the authors claim arise from autosomal misregulation. Phenotypes of misregulated patterning genes are presented as proof, but disruption of several genes produces similar malformations of the developmentally sensitive wing margin.

Do the autosomal fly HAS identified in this study bind CLAMP?

Does a mouse adapter protein localize at sites of MSL2 recruitment at developmental genes?

Valsecchi et al.
NCOMMS-18-04434-T
Point by point response to reviewers

We thank the reviewers for their very constructive comments. In addition to the point by point response to their comments, we provide a short overview of the additional experiments and changes provided in this revised version.

1: Expression analyses in wing discs, larval brains, salivary glands and S2 cells

One of the major highlights of our study is that MSL2 binds and regulates developmental genes on autosomes. One of the comments raised by both reviewers was why they were not scored before and whether these binding sites overlap with previously mapped autosomal sites. We now provide further screenshots (Suppl. Fig. 2a) as well as qPCR expression analyses (Fig. 2g) showing that autosomal MSL2 target genes are neither expressed in S2 cells nor in salivary glands (the two tissues, where MSL2 binding has been previously studied). This provides an explanation, why they were not scored before. Our study therefore brings forward the important concept that the MSL complex may have distinct targets in different tissues, which play an important role during development.

2: Further analyses and comparisons of our ChIP-seq data

Following the requests of the reviewers, we have provided further analyses of our data including genome-browser snapshots and comparisons with previously published ChIP-seq datasets (Fig. 2a and Suppl. Fig. 2a).

3. Further characterization of the MSL2tg transgene

We now provide polytene squashes, immunofluorescence and expression analyses supporting the view that MSL2tg in males reflects endogenous MSL2 (Suppl. Fig. 1c-e and Suppl. Fig. 4b).

4: Characterization of developmental phenotypes upon MSL RNAi

We had shown that wing phenotypes can be triggered upon ectopic expression of MSL2tg in females and *msl-2* RNAi in males. We have now strengthened the analyses in males by reporting additional phenotypes (Fig. 4b-d). We have included *so-Gal4* resulting in eye defects, which supports that the developmental impact is more general and not wing-specific. As suggested by reviewer#1, we have also added RNAi data for other MSL complex members and factors involved in dosage compensation.

5: Separating X-linked from autosomal function

Upon revision, we have provided expression analyses of *hh-Gal4 / UAS-msl-2RNAi* flies, as those male flies survive to adult stages and allowed us to analyze expression differences in male wing discs (Fig. 4f). We are excited to report, that autosomal target genes are regulated by MSL2, while not collectively affecting the X chromosome under such conditions. Together with the ChIP data this supports that MSL2 indeed has a more direct regulatory role on those genes in males.

6: Clarification of data provided in supplementary information

We realized that some of the data presented in the supplementary information was missed by the reviewers. Therefore, in order to ease navigating through the supplementary data, we have now removed the separate Supplementary Method File and put all the respective information in the Materials & Methods section.

Furthermore, we present our data in 5 instead of 4 main Figures. Following the suggestion of reviewer#1, we have also revised the information provided in the figure legends.

The following Figure panels have been **added (new data / analyses)** or **modified** in the revised version of the paper:

Main Figures

- Fig 1a: Added size of Drosophila chromosomal arms below MSL2tg peaks
Fig 1b: Adjusted scalings, exchanged/added snapshots
Fig 2a: Added CLAMP ChIP-seq
Fig 2f: Adjusted scalings, added more snapshots / genes
Fig 2g: Expression analyses of MSL2tg autosomal targets in different tissues
Fig 3a/b: Added *Bx* expression analysis (previously in Supplementary Data)
- Fig 3: Moved panels c-f to the new Figure 4
Fig 4: Data previously in Figure 3, new data added
- Fig 4b: Phenotype using *so-Gal4 / msl-2RNAi*
Fig 4c: Added phenotypes upon *msl-2RNAi* with *wg-Gal4* and *ap-Gal4*
Fig 4d: Table with analyses of phenotypes using RNAi of MSLc members
Fig 4f: Expression analyses in males using 2 different *msl-2* RNAi lines
Fig 4g: Expression analyses in females upon ectopic expression of *msl-2*
Fig 4h: IFs of the X chromosomal territory in male *msl-2* RNAi wing discs
- Fig 5: Previously Figure 4, no changes

Supplementary Figures

- SFig 1b: Added p-values
SFig 1c: Expression analyses supporting that MSL2tg is functional
SFig 1d/e: Polytene squashes of MSL2tg in males and females
SFig 2a: Snapshots comparing MSL2tg ChIP-seq with other datasets
SFig 2b: CLAMP ChIP-seq at HAS
SFig 2e: Expression analysis in different tissues (ctrl for Fig 2g)
SFig 2f: Expression analysis in *mle* null mutant females
SFig 3a: Expression levels now reported relative to males and merged
SFig 3b: Expression analyses of *Bx* and *N* in male and female larval brains
SFig 3c: H4K16ac ChIP-seq snapshots of *Bx*, *N*, *Klp3a*, *Ucp4a*
SFig 3d/e: Removed GFP staining
SFig 3f: Added *CG5254*, *Pfk*, *RpL32*, figure display merged into 1 panel
SFig 3g: Expression analysis in *tub-Gal4 / UAS-msl-2::3Flag* males
SFig 4a: Added higher resolution pictures to visualize the chrX territory
SFig 4b: IFs of the X chromosomal territory in *wild-type* and MSL2tg wing discs
SFig 4e: Added higher resolution pictures of *hh-Gal4/UAS-msl-2* females
SFig 4f: Expression analysis in male *msl-2* RNAi wing discs (ctrl for Fig 4f)
SFig 4g: IFs of the X chromosomal territory in male *msl-2* RNAi wing discs
- SFig 5: Previously Suppl. Figure 4, no changes

Reviewer #1

The current manuscript from Valsecchi et al. presents the novel and intriguing finding that MSL2 (or a MSL complex lacking MLE) in addition to the function on the X-chromosome also targets dosage sensitive autosomal genes involved in patterning and morphogenesis. MSL2 is only expressed in males and changes in amounts of the MSL-complex in males will have dramatic consequences on expression of X-chromosomal genes. This leads to an apparent challenge to distinguish between the direct effects on these autosomal targets and the indirect effects caused by a deregulated X-chromosome. The author claims that the precise regulation of these genes by MSL2 is required for proper development of the fly wing and provide support that MSL2 mediated regulation of this set of genes is conserved in evolution. The paper is based on a large amount of data and the presented results are of general importance, relevance and interest. In particular it provides an intriguing hypothesis on the origin of the MSL2 mediated dosage compensation system and how this system may have been co-opted for dosage compensation during evolution of sex chromosomes.

We thank the reviewer for considering our manuscript as novel and intriguing, while pointing out the general importance, relevance and interest of our study. We have now provided further experiments and controls to strengthen our claims (see below).

CRITIQUE my main concerns are whether the used MSL2tg construct reproduce endogenous MSL2 expression (point 2 below) and whether the statement “that the precise regulation of these genes by MSL2 is required for proper development of the fly wing” is supported by the presented data (point 5 below):

1. MINOR: It is stated in the paper, both in the abstract and the discussion, that the MSLc mediates two-fold upregulation of the single male chromosome. Although often claimed this statement, to my knowledge, still lacks experimental support. The male X-chromosome is upregulated two-fold but the MSLc mediates only part of this, as far as been experimentally shown, see e.g. Hamada et al. 2005, Straub et al. 2008, Zhang et al. 2010, Deng et al. 2009 as well as from the authors lab Vaquerizas et al. 2013. Intriguing recent theoretical arguments state that relation between e.g. RNP2 and mRNA expression is non-linear (Dasmeh et al. 2017). This might turn out to be correct but at this point this needs further experimental support.

We thank the reviewer for raising this point. We completely agree that the precise extent and mechanisms operating on a transcriptional and posttranscriptional level during dosage compensation still have to be worked out. Obviously, lack of MSL-mediated dosage compensation will also lead to secondary responses making this a challenging undertaking. We have carefully rephrased our manuscript accordingly and added the word „approximately“, where necessary.

2. MAJOR: The presented line of arguments is highly dependent on the assumption that MSL2tg (UAS-msl2.FLAG driven by tub.GAL4) reproduce the endogenous expression of MSL2 and the endogenous binding pattern. I am very surprised by the low X-specificity presented in Figure 1a and I would therefore suggest much stronger support that MSL2tg (UAS-msl2.FLAG driven by tub.GAL4) is a reliable proxy for endogenous MSL2 (in males).

Regarding Fig. 1a and “low” specificity: Considering the sizes of the chromosomal arms in *Drosophila*, the MSL2tg ChIP-seq peaks are in fact overrepresented on the X: 848 peaks are found on the X (33.5%) and 1684 on autosomes (66.5%). If peaks were to occur with equal frequency on all chromosomes, one would expect 17.5% peaks on X (23 mb) and 82.5% on autosomes (108 mb). For clarification, we have added the genome size below the panel (Fig. 1a) and the result of the Fisher’s exact test in the text and Figure legend (p-value for overrepresentation of X-linked peaks < 2.2e-16).

Fig. 1c shows that the binding of MSL2tg on HAS (chrX) is in excellent agreement with previously published data (data from Straub et al. 2013, ChIP of endogenous MSL2 in S2 cells). We apologize, if this was not clear and we have now also indicated the reference in the figure legends. As suggested, we have now added

screenshots on X and autosomes in comparison with the datasets of Straub et al. (2013) and Quinn et al. (2016) in Suppl. Fig. 2a.

In the conditions we use, MSL2tg fully rescues lethality of *msl-2²²⁷/ msl-2^{km}* null mutant males demonstrating that this transgene is functional (Suppl. Fig. 1b, compare Bar 4 in the left panel with Bar 4 in the respective cross with the *wild-type* Oregon R strain on the right). The ChIPs were performed in the null mutant genetic background, where MSL2tg is the only source of MSL2. We have carefully chosen the Gal4 driver and temperature (25°C) for expression, as weaker Gal4 drivers or lower temperatures (18°C) did not fully rescue lethality.

To further strengthen the claim that MSL2tg (tub-Gal4, 25°C) reflects endogenous MSL2 in males, we have now also provided expression analyses using qPCR (Suppl. Fig. 1c). This revealed that the assayed genes in *msl-2* null mutants can be rescued to approximately normal levels by providing MSL2tg.

We have also added the polytene squashes suggested by the reviewer (Suppl. Fig. 1d, see below).

To further back this up, we have performed immunofluorescence in wing discs and observe that MSL2tg in males localizes to the typical H4K16ac positive territory. As already reported, this territory does not form in females (Suppl. Fig. 4e). We think that the MSL2tg state in females represents an “intermediate” between a fully dosage compensated male and a normal female (also see below).

Regarding the concern that MSL2tg binding to autosomal targets is an artifact of using a transgene: We can largely exclude that, based on the following controls:

- Suppl. Fig. 1a demonstrates that MSL2tg protein levels accumulate to nearly identical levels compared to endogenous MSL2 in males (lane 4 vs lane 1 in the panel showing endogenous MSL2; also see Villa et al., Mol Cell 2012 and Hallaceli et al., Mol Cell 2012, regarding homeostasis of MSL2 levels).
- We can also exclude that the autosomal sites are an artifact of the FLAG antibody binding to another protein, as we provide an untagged control, which was processed in parallel with the MSL2tg ChIPs (Fig. 1-2 and Suppl. Fig. 1-2).
- Suppl. Fig. 2g-j demonstrate that autosomal binding sites can also be observed with a fully functional, endogenously (CRISPR)-tagged *msl-2::3HA* line, that faithfully localizes to the X in polytene squashes.
- If binding at autosomal sites would occur “aberrantly” due to ectopic expression of MSL2tg one could expect a misregulation of these genes in males and presumably the appearance of a wing phenotype - however both of these things do not happen (Suppl. Fig. 3g and Fig. 4a).
- However, if the binding at these genes is indeed real, one would expect that autosomal target genes, which lose *msl-2* mediated regulation, would be downregulated upon *msl-2RNAi* in males (while displaying a phenotype, Fig. 4c). We provide these expression analyses upon revision and are happy to report such downregulation in males, when using tissue-specific drivers.

Regarding the concern that these autosomal sites and/or regulation were not detected before, neither in ChIPs from S2 cells, salivary glands nor in polytene squashes:

Polytene squashes and confocal microscopy have only limited resolution and sensitivity. This is important to consider with regards to scoring local binding on individual genes (MSL2tg binding on autosomal targets, e.g. *wg*), in comparison with broad, spreading-dependent binding (e.g. MSL binding on the X (Fig. 1 and 2)).

As our identified targets play a role in developmental processes, we have now also added expression analyses and further comparisons with previously published ChIP data in Fig. 2 and Suppl. Fig. 2. We find that autosomal MSL2 target genes are neither expressed in S2 cells nor in salivary glands (Fig. 2g). Given that the MSL complex operates on active genes (e.g. Kind et al. 2007, Larschan et al. 2007, Alekseyenko et al., 2012), this provides a possible explanation, why these targets were not scored before.

In agreement with this, we find the CLAMP adapter protein (Kuzu G. et al., 2016) enriched on both HAS and cluster 1 autosomal sites (Fig. 2a), which both contain the characteristic GAGA-rich motif. Of note, this CLAMP ChIP dataset has been generated from whole larvae and hence, can be more directly compared with our MSL2tg data.

Taken together, we hope that these controls provided in our manuscript, together with the phenotypic data showing MSL-mediated regulation of these genes, now further convinces the reviewer that the binding at autosomal genes is real: It indeed occurs in males, but only in tissues expressing developmental regulatory genes.

Examples of data to better support this assumption:

a. Polytene chromosome stainings with this transgene. Is it as X-specific as the endogenous MSL2? Please, also include polytene stainings for females expressing MSL2tg since these are viable to adult stage in contrast to previously published msl2 transgenes expressed in females.

We thank the reviewer for this suggestion. Upon revision, we have now performed these experiments, which show a pattern that reflects the expected localization of MSL2tg and the other MSL complex members on the male X. Please note that the FLAG-epitope sequence (DYKDDDDK) contains Lysines and is therefore not ideally suited for immunofluorescence. Therefore, the MSL2tg squashes in females, which express significantly lower MSL2tg levels than males (Suppl. Fig. 1a), showed high levels of background. However, MSL1 was properly targeted to the X supporting faithful targeting of the MSLc by MSL2tg in females as well. Please also refer to our responses regarding lethality and the induction of X chromosomal territory in MSL2tg females.

b. How comes (if I understand it correct) that expressing MSL2tg in females doesn't cause lethality?

Suppl. Fig. 1b shows a reduction in viability of around 30%, to which we have referred to as "mild lethality" (p.10 of our manuscript). We have now added p-values to make this point clearer. Kelley et al. (1995, Cell) reported that "Many of the H83M transgenic lines displayed a dominant female-specific developmental delay (data not shown). Within this group, a subset of lines had decreased female viability and severely impaired female fertility." We have never been able to observe extensive lethality in females with our *msl-2* transgene. We tested various approaches such as using higher temperatures, other strong Gal4-drivers or by recombining our allele on chr3 with alleles on chr2 (Hudry et al. Nature, 2016). We have also contacted the lab of Irene Miguel-Aliaga who generated the aforementioned transgenic *msl-2* lines. Like us, they have also observed a reduction in viability (around 50%, personal communication) but not complete lethality. Therefore, there must be something special about the nature of this particular *hsp83-msl-2* line characterized in the 1995 publication, possibly the fact it has two insertions of *msl-2* (reported in Lyman et al., Genetics 1997). This certainly requires further clarification by the field.

c. As presented, the ChIP profiles are not fully convincing. Please show profiles over larger regions with comparisons of ChIP profiles for MSL2tg with endogenous MSL2 (e.g. from Straub et al.) or endogenous MSL1, MSL3 or MOF. Show profiles for e.g. 1Mb of the X-chromosome and 1Mb from an autosome including some of the autosomal genes of interest. Use the same scale for X and autosomes for each profile.

As requested, we have provided more snapshots in comparison with the data published by Straub et al. (Suppl. Fig. 2a) in addition to the ones already provided in Fig. 1 and 2 (Heatmaps referred to as “MSL2 ChIP S2 cells”). We now also added the reference to Straub et al. (Genome Res. 2013) as requested in the Figure legends.

In the following panel, we also provide screenshots comparing our dataset with the MSL2 profiles from Figueireido et al. (PLOS Genetics, 2014) generated from salivary glands. This dataset was sequenced on a different NGS platform (AB SOLiD 5500xl). It is therefore more difficult to compare to our profile and/or the one published by Straub et al. (2013), both in terms of resolution and enrichment. However, all profiles consistently show MSL2 target sites on the X are robust and observed by different labs and different techniques and resolutions.

3. Presenting ChIP profiles (and ChIRP profiles)

a. In Figure 1b, 2f, 4a please use the same scale on the y-axis (for one specific experiment) as much as possible. For example in Figure 1b it is impossible to judge if the roX2 peaks in *wg* and *en* are significant. If different scales are required this needs to be clearly pointed out in e.g. the figure legend. Please also ensure that the information on experiments are included in the figure legend. For example, I assume that roX2 in figure 1b refers to the roX2 ChIRP from Quinn et al 2016 but this may be wrong and it is not indicated in the legend.

We apologize, if this was not clear (see the point above). Yes, the roX ChIRP data is from Quinn et al. (2016). We have now clarified this as requested in the figure legends.

Regarding the scaling:

1) The roX enrichment by ChIRP is indeed much higher on X-linked sites compared with autosomal sites. We had previously mentioned this on p.16 of our manuscript and now indicated this as well on p.7. We think this low level of roX enrichment is not random, given that we don't find it in Cluster 2 and 3 (Fig. 2a). We think that it is worthwhile pointing out this interesting finding to the field, for which the different scaling was required. For the revised manuscript, we have now applied the same scaling on all autosomal sites in Fig. 1b.

2) Enrichments scored in different ChIP-seq experiments depend on many factors, including the antibody *per se*, the IP conditions, library preparation or the sequencing technology. Therefore, different proteins and profiles have different enrichment levels on different loci and different scalings may be applied. This is common practice and is also done for example by Straub et al. (2013). We have consistently used the

same scales for samples that were processed together, e.g. MSL2tg replicates together with untagged controls (Fig. 1b, Fig. 2f).

Following the reviewer's advice, we have now carefully revised all the scalings and provided further information on scaling in the figure legends, where necessary.

b. Concerning the y-axis scale for ChIP and ChIRP profiles – what is actually shown? I understand it shows enrichment and not in a log2 scale but if so how comes the negative values?

The y-scale represents the difference of sequencing depth normalized ChIP versus Input (see Materials & Methods, data was processed according to ModEncode guidelines). Since the normalization scales the total coverage to be 1, a difference of 0 means that both ChIP and input had about the same number of normalized reads while values over zero mean an enrichment of ChIP over Input and negative values mean an enrichment of Input over ChIP (=background).

c. To be able to judge the significance of the peaks shown in ChIP and ChIRP profiles please provide figures on zoomed out browser views on e.g. 1Mb from the X-chromosome and from an autosome with the same y-axis scale for comparison.

We have provided the requested screenshots (Suppl. Fig. 2a). We would like to emphasize that the statistical significance of the enriched sites is provided by peak calling, for which we have used MACS2 with a q-value of 0.001 (Zhang, Y. et al. Genome biology 9, 2008). Moreover, the peaks are not found in untagged controls.

d. ChIP qPCR results to support are shown in SFig 2d. Please show this as “% of input”. The y-axis scale “fold change over non targets” is not very informative on the ChIP quality and that HAS and “target genes” are enriched just 2-4 fold more than the control amplicons ent2 and CG15011 is not in line with previously published enrichments.

In the experiment displayed in Suppl. Fig. 2d, we have chosen the same y-axis scaling as Straub et al. (2013), where Figures 1D and 1E show a roughly 5 to 10 fold enrichment of MSLs over non-targets. Prestel et al. report a roughly 4-fold enrichment of MSL2 on the X-linked *armadillo* over non-targets (Figure 2E, Mol Cell 2010). We are also aware of MSL2 ChIP data published by the Larsson (Figueiredo et al., PLoS Genetics 2014) and Becker labs (Straub et al., PLoS Genetics 2008), but both do not provide a qPCR quantification for MSL2. The MSL2 ChIP in Figure 5A of Larschan et al. (Mol Cell, 2007) is presented as “% IP for MSL2 normalized to Input and PKA” and bars for HAS reach to 350%. This particular scaling is unclear to us. In Larschan et al. (PLOS Genetics, 2012) MSL2 ChIP enrichments are reported by setting the wild-type enrichment to 100%.

However, for transparency we are happy to provide the following figure showing % of Input enrichment for the reviewer:

To be consistent with the previously published MSL2 ChIP data mentioned above, we though prefer to keep the display as it is (Suppl. Fig. 2g), so it is indeed comparable.

4. Classification of targeted genes: 49 genes are classified as “cluster 1”, 176 as cluster 2 and 283 as “cluster 3” in total 508 genes.

a. The number of targeted genes (508) makes it important to clarify the logics in which “example genes” that are shown. The genes *wg*, *en*, *ap*, *so*, *mirr* and *opa* are all classified as cluster 1 while *vg* is classified as cluster 3. Please state this in the text and explain why including these specific genes.

The three groups / clusters have been generated using an unsupervised k-means clustering algorithm. As suggested by the reviewer, we have now also stated this in the text. For validation, we have selected the aforementioned genes based on evolutionary conservation (hence more genes of cluster 1, Fig. 5g), function in morphogenesis and reported connection to the phenotypes described in our manuscript. From this point of view, there are many more extremely interesting genes in our lists, which we and the dosage compensation field will certainly expand on in the future. Based on the reviewer’s suggestion we have now ensured that our phrasing regarding the choice of example genes is clear in the text.

5. MAJOR: One important statement in the paper is that the precise regulation of these genes by MSL2 (the autosomal ones like *vg*, *wg*, *en* etc.) is required for proper development of the fly wing. This is a novel important finding but also a statement with some caveats.

a. Figure 3c shows that overexpression of *msl2* in females causes a wing defect (but not lethality). Maybe I am wrong in this but to me an induction of MSL2 in females should increase expression levels on many genes (thousands) and it is impossible to distinguish direct effects from the suggested target genes from indirect effects from e.g. X-chromosomal genes.

As discussed above, we observe partial lethality by ectopic expression of MSL2tg in females. We have not performed extensive RNA-seq experiments to globally assess this point. However, we can already state, that this does not result in a promiscuous upregulation of all genes on X or autosomes (Fig. 3, Suppl. Fig. 3, e.g. the X-linked *Ucp4a*, *CG5254* or *roX1* are not affected upon ectopic expression, although they are robust, X-linked MSL-targets in males (Suppl. Fig. 1c)). Given that there is incomplete formation of a X chromosomal territory (Suppl. Fig. 3 and 4), we think that expressing MSL2tg in females in our conditions results in an “intermediate” state between a fully dosage-compensated male and a normal female without DC. Analysis of this “intermediate” state (together with the *hh-Gal4/msl-2* RNAi experiments in males) allows us to at least partially separate global, X-linked function from local gene-by-gene effects. Nevertheless, this will never exclude that some X-linked genes are involved (for example, some of the PionX genes that seem to behave like autosomal targets), and we have therefore carefully rephrased our text where necessary (also see below).

b. Figure 3e shows wing defects in males when *msl2* is depleted through RNAi. Again, *msl2* depletion should cause a changed expression levels of many genes and it is not possible to distinguish direct from indirect effect (and X-chromosome targets from autosomal targets). However, to control for this the authors show that depletion of *mle* will not give the same phenotype. This becomes a critical experiment and in my opinion much more support is needed here. As I understand the paper the prediction would be that depletion of *msl2*, *msl1*, *msl3* or *mof* should generate the wing phenotype while depletion of *mle* should not under the assumption that the RNAi for all these genes is similar in efficiency.

We have provided additional RNAi experiments in Fig. 4.

To address the issue of RNAi efficiency, we have verified our results concerning *msl-2* with an independent line and are happy to report consistent results. We have also attempted to provide analyses of an independent RNAi line for *mle*. Unfortunately, this line did not display male-specific lethality with *tub-Gal4* and qPCR analyses revealed that there was no depletion of *mle*.

Regarding other MSL members: We find that we can trigger wing phenotypes with *hh-Gal4 / msl-1* RNAi, whereas *hh-Gal4 / msl-3* and *mof* RNAi - interestingly - resulted in male-specific lethality. Note that all these lines were strong enough to result in male-specific lethality using *tub-Gal4*, indicating that they are specific and have comparable efficiencies. *Pof* RNAi did not result in a phenotype (Johansson et al., Mol Cell Biol. 2012) supporting our conclusion that these effects are MSL-specific.

My question is how the X-chromosomal effect can be convincingly separated from the "autosomal target effect"?

We have now provided expression analyses in *hh-Gal4 / msl-2RNAi* male wing discs using two different lines. We are happy to report that autosomal target genes are downregulated upon *msl-2RNAi* in males, while the X was not collectively misregulated in these conditions. Of course, this does not exclude that some alterations of X-linked genes are found in the misregulated gene expression state eventually resulting in the wing phenotype.

Yet, it clearly strengthens our point that the regulation of these autosomal MSL2 targets is direct:

- These genes are bound in ChIP
- They are misregulated in males (downregulated upon RNAi, 2 independent lines) and females (upregulated upon ectopic expression, 2 independent Gal4 drivers)
- In both scenarios these gene expression changes occur independently of *global* effects on the X as assessed by
 - no collective upregulation or gain of territory in females
 - no collective X chromosomal territory loss in the two different RNAi lines (31627, 35390). Yet, expression changes on autosomal targets are consistent and more pronounced than most X-linked genes in both lines.

However, we agree with the reviewer's comment that completely separating X from autosomal function is presumably impossible. We have now adjusted our wording in the text appropriately, in addition to reporting these exciting results.

ADDITIONAL COMMENTS:

6. On page 3 line 5 the abbreviation "DC" is used for the first time but without explanation, please correct.

We have corrected this.

7. *Just out of curiosity – why was a ChIP-seq using the MSL2 antibody on wildtype males not included to assess how well MSL2tg reproduce endogenous expression in males?*

Unfortunately, the MSL2 antibody (sc-32459) is not produced anymore by Santa Cruz (personal communication), so we couldn't perform the extensive optimizations and analyses required for ChIP with these limited amounts that are remaining in our lab.

8. *Page 6 line 11: “Remarkably, the H4K16ac pattern upon deletion of msl-2 in males was looked essentially identical to the one in wild-type females”. Why is this “Remarkable”? Isn't it what we would expect?*

Given the challenges in collecting *msl-2* null mutant males, this ChIP profile - from a methodological point of view - represents a highlight in our manuscript. However, since the pattern indeed looks as expected we have removed this statement.

9. *Page 6 last sentence: “indicating that the global regulation of the X is presumably invariant in different tissues”. Please explain this statement in relation to the claim that “accessibility of these sites is cell-type specific”.*

We have rephrased these sentences.

10. *Please be consistent in naming profiles. For example, in Figure 1 the MSL2tg profile is named MSL2tg but in Figure 2f the profile is named MSL2 and in Figure 3 dmMSL2*

We have corrected this.

11. *Please review and correct genetic nomenclature. For example in Fig 3f the genotype is stated as hh-GAL4/UAS-GFP, UAS-msl2::Flag3. I don't understand this. Was the UAS-GFP transgene recombined into the UAS-msl2::Flag3 chromosome and if so which UAS-GFP was used and was this setting used in other experiments as well?*

We thank the reviewer for pointing this out and have corrected the nomenclature.

12. *Sfig1a: MSL3 shows a double band in the control but not in the other genotypes – why?*

At the moment, we don't know why this MSL3 doublet appears in OregonR flies but not in our experimental crosses. The protein extracts presented in the figure were prepared at the same time, while using the same buffers. We can only speculate that an unspecific protein being only present in OregonR causes the additional band.

13. *Sfig1c: What are the H4K16ac and roX2 profiles – which experiments?*

The datasets that are used in our figures are described in Suppl. Table 1: H4K16ac ChIP-seq was generated in this study, roX2 ChIRP by Quinn et al. (2016).

14. *Sfig3d: Why do we see GFP staining in the 3d panel? According to the genotype stated there is no UAS-GFP in this line.*

We completely agree with the reviewer that the Rabbit polyclonal GFP antibody used in these IFs causes quite a lot of background. In fact, this can also be seen in the IFs from *UAS-GFP ; ; hh-Gal4 / UAS-msl-2::3Flag* wing discs, where we have used this particular antibody (Fig. 4i). Here, GFP can also be detected in the anterior part, where *hh* is not expressed (Basler & Struhl, Nature 1994, Tanimoto et al. Mol Cell 2000). The actual point of the panel in Suppl. Fig. 3d was to show the specificity of the FLAG staining. We now realized that the relatively unspecific GFP staining may

raise more questions than answers and have therefore decided to remove the respective staining.

15. Supplementary figure 3e is supposed to show that MSL2tg does not localize to a typical H4K16ac-positive territory characteristic for the male X. The resolution of the provided figure is not enough to draw this conclusion.

We have provided higher resolution pictures to support this claim.

Reviewer #2

This study starts with the very interesting premise that MSL2 orchestrates developmental gene regulation on the autosomes of flies, and that regulation of developmental genes by MSL2 is evolutionarily conserved in mammals. The normal function of MSL orthologs in mammals is relatively poorly studied, making this an advance. This idea could explain the binding of MSL proteins to a number of well-known autosomal sites of flies. However, many of the conclusions are flawed by potentially qualifying technical issues. Furthermore, aspects of the current findings disagree with prior studies in trivial or significant ways. These differences are not acknowledged or discussed.

The story in mouse cells is interesting and parallels with flies intriguing. While the foundational premise of this work is interesting (and may ultimately be biologically important and exciting) the current manuscript is reaching for a story that is not quite there yet.

We thank the reviewer for considering our manuscript as biologically important and exciting, while particularly pointing out that our mammalian data represents an advance. However, we respectfully disagree that our data “are flawed”. In fact, we had already provided many replicates as well as technical and biological controls in our initial manuscript. Due to space limitations, we had put some of these controls in the Supplementary Figures and we can therefore only assume that for this reason our point has not always been entirely clear. While clarifying the specific points below, we have now provided even more controls and experiments to strengthen our claims.

Specific criticisms

Autosomal MSL binding sites have been previously studied. No effort is made to determine if sites currently identified correspond to ones mapped previously.

In Fig. 1 and 2, we show previously published data for MSL1 (Chlamydas et al., 2016, Salivary glands), MSL2 (Straub et al. 2013, S2 cells), roX (Quinn et al., 2016, whole larvae) in comparison with our profiles for MLE and MSL2tg. We apologize, if this was not clear and have now added the respective information also in the Figure legends. We have also provided further screenshots in comparison with Straub et al. (2013) in Suppl. Fig. 2a.

In addition to the aforementioned MSL members, we have now also analyzed CLAMP ChIP-seq (Kuzu et al., 2016, Fig. 2a), which is in great support of our conclusions.

We are also happy to provide the following figure for the reviewer, which compares our dataset with the salivary gland MSL2 ChIP-seq profiles from Figueireido et al. (PLOS Genetics, 2014). Note that this dataset was sequenced on a different NGS platform (AB SOLiD 5500xl). Therefore, it is more difficult to compare to our profile and/or the one published by Straub et al. (2013).

Regarding correlations to polytene squashes obtained from salivary glands see below.

The discovery that ectopic MSL2 expression elicits inappropriate dosage compensation in females is over 20 years old, but is presented as novel (p.5).

Because there are varying degrees of lethality observed with different *msl-2* transgenes (Hudry et al., Nature 2016, Kelley et al., Cell, 1995), our intention was to simply report on what we observe in our newly generated MSL2tg allele. To the best of our knowledge the MSL2tg ChIP-seq in females generated by us is the first *-seq profile in females upon ectopic expression *in vivo* (L3 larvae). We think that it's important to relate our extensive characterizations of MSL2tg to what has been previously observed with other alleles (e.g. *hsp83::msl-2* in the paper by Kelley et al. 1995, Cell). In our manuscript, the term "novel" is used solely with regards to the autosomal sites identified in ChIP (p.7). However, we understand the reviewers' concern and have now removed the sentence on p.10/11 regarding the induction of lethality.

The authors maintain that MSL2 binds autosomal chromatin, but MLE does not. This contradicts previous findings that MLE is found throughout the genome of females, and only becomes restricted to the X in males (Genetics 172, 963 and references therein). Furthermore, MLE is tethered to chromatin by RNA. MNase is used for chromatin preparation in this study, selectively releasing MLE.

In our ChIP protocol, *Drosophila* larvae are fixed before the isolation of nuclei and MNase treatment (see methods). The MLE ChIP-seq datasets in males and females have been processed in parallel and we are able to score MLE association with the X chromosome in males (Fig. 1a and c). Hence, our protocol faithfully captures RNase-sensitive MLE binding sites on the male X (Ilik et al., 2013, Richter et al., 1996). Moreover, our MLE ChIP is in great agreement with the data published by the Becker lab in male S2 cells (Straub et al. Genome Research, 2013), which uses sonication instead of MNase for chromatin fragmentation. As discussed in our manuscript, it is possible that the lower levels of roX at autosomal sites are not sufficient to result in MLE recruitment in males.

Regarding the autosomal MLE sites in females that have been reported by Kotlikova et al. (2006), Bhadra et al. (1999) and Kuroda et al. (1991): One possibility is that higher concentrations of formaldehyde were used for fixation in these stainings (typically 3-4%, exact concentrations are not mentioned in the Materials & Methods) than in our ChIPs (1% for MLE), which allows to score more transient or indirect interactions. Moreover, polytene squashes and confocal microscopy have different resolution and sensitivity compared to ChIP. This is important to consider with regards to scoring local binding on individual genes (e.g. MSL2tg binding on autosomal targets, e.g. *wg*), in comparison with broad, spreading-dependent binding (e.g. MSL binding on the X (Fig. 1 and 2)) and the aforementioned MLE sites on female autosomes in salivary glands.

Furthermore, if autosomal MLE binding in females would be functionally relevant and/or overlapping with regards to our targets, one would expect a misregulation of such genes in female *mle* null mutants. To address this, we have performed qPCR analyses but do not find any evidence for up- or downregulation of our identified MSL2 target genes in *mle* null mutant females (Suppl. Fig. 2f). Moreover, MLE RNAi does not result in a wing-phenotype in females, neither do MLE null mutant females display a wing-phenotype or any other obvious developmental abnormalities. However, this would be the expectation, if the female binding is functionally overlapping with MSL2tg binding sites that are linked to such developmental alterations. Last, we think that it is generally unlikely that the salivary gland-sites and

our MSL2 ChIP peaks overlap, because the autosomal, developmental-regulatory MSL2 target genes are largely not expressed in salivary glands (Fig. 2g).

We have adjusted our text according to these results.

Bx is indeed dosage compensated. This gene is responsible for characteristic wing notching in triplo-X females, as well as in females that ectopically express MSL2 (Genetics 183, 811). Increased expression of Bx is almost certainly the source of the defects that the authors claim arise from autosomal misregulation. Phenotypes of misregulated patterning genes are presented as proof, but disruption of several genes produces similar malformations of the developmentally sensitive wing margin.

We thank the reviewer for raising this interesting point. The Genetics 183, 811 publication and other references concerning *Bx* had not escaped our attention and for this reason, we had already provided expression analyses of *Bx* in our initial manuscript. To make our point clearer, we have now put the *Bx* qPCR in the main figure and added H4K16ac ChIP-seq snapshots of males in comparison with females in Suppl. Fig. 3. We also added expression analyses of *Bx* in larval brains. Most importantly, we find that *Bx* levels are not changed upon ectopic expression of MSL2 in females (Fig. 3).

Our conclusion from this data is:

Bx and *N* are escape genes, which are not subjected to MSL-mediated dosage compensation and hence are lower expressed in males compared to females. In our view, this speaks against the hypothesis that misregulation of *Bx* is the primary cause for the wing phenotypes observed in our paper. Of course, it is possible that these genes participate in some way in the complex feedback loops operating during wing morphogenesis.

Regarding the references mentioned by the reviewer:

Sun and Birchler (Cytogenetic and Genome research, 2009, also see the review by Birchler, 2016) report that the majority of X-linked genes in metafemales (3X) equalize to normal levels (2X), whilst autosomal genes change. They also mention that females heterozygous for *Bx1/+* and *Bx2/+* on the X together with the *hsp83::msl-2* transgene show “no evidence of acquiring dosage compensation giving a normal wild type wing phenotype in both cases (data not shown)”. The *Bx* allele used in the interesting study by Menon & Mellor (Genetics, 183, 811) is a duplication of at least 47 genes (17A6-17C7). A wing phenotype is reported in *Bx/+ hsp83::msl-2* females, that is suppressed by *msl-1* mutation. However, given that overexpression of *msl-2* alone already gives a wing phenotype (our study) while taking into account the results reported by Birchler it seems to us that the roles of *Bx* are not entirely clear. Unfortunately, there are no molecular analyses presented in these papers (ChIP or expression analyses in males and females) that would allow us to put our results in context with these studies.

Taken together, we completely agree with the reviewer that the complex interplay between autosomal and X-linked genes, some of which display MSL-mediated dosage compensation, while others escape (*N*, *Bx*) is very interesting. However, we hope that the reviewer understands that in the interest of not losing the main message of our manuscript - the evolutionary conserved regulation of developmental regulatory genes by MSL2 - and the given space, we are unable to specifically discuss the role of *Bx* and the aforementioned papers in much greater detail.

Do the autosomal fly HAS identified in this study bind CLAMP?

We have provided this analysis upon revision and are happy to report, that cluster 1 autosomal sites (GAGA-rich motif) are also enriched for CLAMP (Fig. 2).

Does a mouse adapter protein localize at sites of MSL2 recruitment at developmental genes?

We thank the reviewer for raising this point. In agreement with what we have described earlier in Chelmicki et al. (Elife, 2014), we find YY1 enrichment at a subset of MSL-targets in mammalian cells. However, a more extensive characterization of the mechanisms involved in MSL recruitment in mammalian cells in comparison with *Drosophila* goes far beyond the scope of this manuscript.

Reviewers' comments:

Reviewer #1 (Remarks to the Author):

The authors have responded to my initial critiques and concerns, moderated and corrected the text and figures. I find the revised version convincing. Having read through the paper again also in the light of the second reviewer and the responses, I recommend acceptance of this highly intriguing story.

Reviewer #2 (Remarks to the Author):

This manuscript reports a number of interesting correlations, but fails to nail down causation. A primary concern remains the quality and consistency of data that has been deployed in support of a rather spectacular story. While the story may well be true, the torrent of data is displayed in a manner that thwarts careful analysis.

The equivalence of a tagged MSL2 transgene and endogenous MSL2 was flagged in the original review but has really not been settled. This matters because autosomal MSL2 binding depends on expression levels. A blot suggests that MSL2tg protein levels are within 3 or 4 fold that of wild type, but Sup. Fig. 1c reveals that the tub-Gal4 driver achieves ~100 fold increase in MSL2 mRNA over that of a fly heterozygous for an MSL2 mutation. I hope that this is a misunderstanding of the figure.

Fold enrichment at binding sites is reported to be comparable between MSL2tg and MSL2-HA tagged at its endogenous locus, but data is presented in a manner that discourages comparison. There are few genes in common between Fig. 1b and Supp Fig. 2 i. The genes that are found in both, roX1 and ap, reveal very different enrichment when detected by MSLtg (27 and 20 fold enrichment) and MSL2-HA (3 and 4 fold enrichment). I am not sure what to make of this as there are technical differences.

A question previously raised was whether autosomal sites identified in the present study using ChIP overlap with those previously mapped cytologically. This question was not addressed. It would be useful to know how closely molecular and cytological approaches harmonize. This requires simply consulting the literature.

A reviewer asked about the presence of CLAMP at autosomal sites. The authors provided documentation of CLAMP at X-linked sites, something that is already well-known, but failed to address the presence or absence of CLAMP at autosomal sites.

Valsecchi et al.
NCOMMS-18-04434-T
Point by point response to reviewers

Reviewer #1 (Remarks to the Author):

The authors have responded to my initial critiques and concerns, moderated and corrected the text and figures. I find the revised version convincing. Having read through the paper again also in the light of the second reviewer and the responses, I recommend acceptance of this highly intriguing story.

We thank Reviewer #1 for his/her positive reply and for considering our manuscript highly intriguing.

Reviewer #2 (Remarks to the Author):

This manuscript reports a number of interesting correlations, but fails to nail down causation. A primary concern remains the quality and consistency of data that has been deployed in support of a rather spectacular story. While the story may well be true, the torrent of data is displayed in a manner that thwarts careful analysis.

In our previous point-by-point response, we had already commented on the issue of causality to Reviewer #1, in particular on how autosomal binding can be separated from X chromosomal effects in *Drosophila*. In the first revision of our manuscript, we have therefore provided expression analyses, showing that autosomal targets are downregulated upon *msl-2^{RNAi}* in males, while the X was not collectively misregulated. Of course, this does not exclude that some alterations of X-linked genes are found in the misregulated gene expression state eventually resulting in the wing phenotype. Yet, it clearly strengthens our point that the regulation of the autosomal MSL2 target genes is direct:

- These genes are bound in ChIP
- They are misregulated in males (downregulated upon RNAi, 2 independent lines) and females (upregulated upon ectopic expression, 2 independent Gal4 drivers)
- In both scenarios these gene expression changes occur independently of global effects on the X as assessed by
 - No collective upregulation or gain of territory in females
 - No collective X chromosomal territory loss in the two different RNAi lines (31627, 35390).
 - Yet, expression changes on autosomal targets are consistent and more pronounced than most X-linked genes in both lines.
- No phenotype upon *mle^{RNAi}* in males, although this line is strong enough to cause male-specific lethality using *tub-Gal4*. This is consistent with lack of MLE enrichment at autosomal sites and hence, the absence of spreading.

However, we agree that completely separating X from autosomal functional is presumably impossible. We had therefore already adjusted our wording in the text appropriately in the first revision of our manuscript, a fact that was also appreciated by Reviewer #1.

Below, we have addressed any remaining specific concerns of Reviewer #2 regarding the quality and consistency of the data. Throughout the manuscript, we have been

careful to display the data in a clear and consistent manner, and do not believe that the manner in which the data are displayed impedes analysis.

The equivalence of a tagged MSL2 transgene and endogenous MSL2 was flagged in the original review but has really not been settled. This matters because autosomal MSL2 binding depends on expression levels. A blot suggests that MSL2tg protein levels are within 3 or 4 fold that of wild type, but Sup. Fig. 1c reveals that the *tub-Gal4* driver achieves ~100 fold increase in MSL2 mRNA over that of a fly heterozygous for an MSL2 mutation. I hope that this is a misunderstanding of the figure.

First of all, we are surprised that the issue of equivalency of MSL2tg expression levels is now raised by Reviewer #2, as the Western blot was already provided in the first submission of our manuscript.

Related to *msl-2* RNA levels: Indeed, *msl-2* RNA levels driven from *tub-Gal4* are much higher than endogenous *msl-2* RNA levels and we have reported these values in Suppl. Fig. 2c to a) be transparent and b) prove that the black bars / samples actually represent MSL2tg expressing larvae. We would like to point out that the transgene (as indicated in the main text, materials and methods and figure legends) only contains the coding sequence (CDS), but not regulatory sequences. It is broadly documented in the literature that RING domain proteins are regulated at a posttranscriptional level by proteasomal degradation (e.g., Soucy et al., Clin. Cancer Research, 2009). In particular, for MSL2 it is well described in studies published by several different labs that homeostasis / protein levels are tightly controlled (e.g., Hallacli et al., Villa et al. Mol Cell, 2012). So, this apparent “inconsistency” between RNA and protein levels is highly consistent with the literature. Please also note that we were not able to rescue male-specific lethality with weaker Gal4 drivers or at lower temperatures.

We would also like to point out the result of the quantification of the Western blot (Suppl. Fig. 1a) to the reviewer. Protein levels in lane 4 (the conditions where the CHIP was performed) are about 1.5-fold relative to the endogenous MSL2 protein in wild-type OregonR male flies (lane 1).

Please also note our response regarding the induction of lethality in females below as well as in our Response to Reviewer #1, from which we conclude that MSL2tg levels (*tub-Gal4*, *UAS-msl-2* at 25°C) are matching very closely to the ones observed from the endogenous promoter.

For clarification, we have copied the response to Reviewer #1 again, which includes a detailed discussion of this issue.

Our earlier response to Reviewer #1 regarding the equivalency of MSL2tg versus MSL2.

In the conditions we use, MSL2tg fully rescues lethality of *msl-2²²⁷/msl-2^{km}* null mutant males demonstrating that this transgene is functional (Suppl. Fig. 1b, compare Bar 4 in the left panel with Bar 4 in the respective cross with the *wild-type* Oregon R strain on the right). The ChIPs were performed in the null mutant genetic background, where MSL2tg is the only source of MSL2. We have carefully chosen the Gal4 driver and temperature (25°C) for expression, as weaker Gal4 drivers or lower temperatures (18°C) did not fully rescue lethality.

To further strengthen the claim that MSL2tg (*tub-Gal4*, 25°C) reflects endogenous MSL2 in males, we have now also provided expression analyses using qPCR (Suppl. Fig. 1c). This revealed that the assayed genes in *msl-2* null mutants can be rescued to approximately normal levels by providing MSL2tg.

We have also added the polytene squashes suggested by the reviewer (Suppl. Fig. 1d, see below).

To further back this up, we have performed immunofluorescence in wing discs and observe that MSL2tg in males localizes to the typical H4K16ac positive territory. As already reported, this territory does not form in females (Suppl. Fig. 4e). We think that the MSL2tg state in females represents an “intermediate” between a fully dosage compensated male and a normal female (also see below).

Regarding the concern that MSL2tg binding to autosomal targets is an artefact of using a transgene: We can largely exclude that, based on the following controls:

- Suppl. Fig. 1a demonstrates that MSL2tg protein levels accumulate to nearly identical levels compared to endogenous MSL2 in males (lane 4 vs lane 1 in the panel showing endogenous MSL2; also see Villa et al., Mol Cell 2012 and Hallaçli et al., Mol Cell 2012, regarding homeostasis of MSL2 levels).
- We can also exclude that the autosomal sites are an artefact of the FLAG antibody binding to another protein, as we provide an untagged control, which was processed in parallel with the MSL2tg ChIPs (Fig. 1-2 and Suppl. Fig. 1-2).
- Suppl. Fig. 2g-j demonstrate that autosomal binding sites can also be observed with a fully functional, endogenously (CRISPR)-tagged *msl-2::3HA* line, that faithfully localizes to the X in polytene squashes.
- If binding at autosomal sites would occur “aberrantly” due to ectopic expression of MSL2tg one could expect a misregulation of these genes in males and presumably the appearance of a wing phenotype - however both of these things do not happen (Suppl. Fig. 3g and Fig. 4a).
- However, if the binding at these genes is indeed real, one would expect that autosomal target genes, which lose *msl-2* mediated regulation, would be downregulated upon *msl-2^{RNAi}* in males (while displaying a phenotype, Fig. 4c). We provide these expression analyses upon revision and are happy to report such downregulation in males, when using tissue-specific drivers.

Regarding the concern that these autosomal sites and/or regulation were not detected before, neither in ChIPs from S2 cells, salivary glands nor in polytene squashes:

Polytene squashes and confocal microscopy have only limited resolution and sensitivity. This is important to consider with regards to scoring local binding on individual genes (MSL2tg binding on autosomal targets, e.g. *wg*), in comparison with broad, spreading-dependent binding (e.g. MSL binding on the X (Fig. 1 and 2)).

As our identified targets play a role in developmental processes, we have now also added expression analyses and further comparisons with previously published ChIP data in Fig. 2 and Suppl. Fig. 2. We find that autosomal MSL2 target genes are neither expressed in S2 cells nor in salivary glands (Fig. 2g). Given that the MSL complex operates on active genes (e.g. Kind et al. 2007, Larschan et al. 2007, Alekseyenko et al., 2012), this provides a possible explanation, why these targets were not scored before.

In agreement with this, we find the CLAMP adapter protein (Kuzu G. et al., 2016) enriched on both HAS and cluster 1 autosomal sites (Fig. 2a), which both contain the characteristic GAGA-rich motif. Of note, this CLAMP ChIP dataset has been generated from whole larvae and hence, can be more directly compared with our MSL2tg data.

Taken together, we hope that these controls provided in our manuscript, together with the phenotypic data showing MSL-mediated regulation of these genes, now further convinces the reviewer that the binding at autosomal genes is real: It indeed occurs in males, but only in tissues expressing developmental regulatory genes.

Fold enrichment at binding sites is reported to be comparable between MSL2tg and MSL2-HA tagged at its endogenous locus, but data is presented in a manner that discourages comparison. There are few genes in common between Fig. 1b and Supp Fig. 2 i. The genes that are found in both, roX1 and ap, reveal very different enrichment when detected by MSLtg (27 and 20 fold enrichment) and MSL2-HA (3 and 4 fold enrichment). I am not sure what to make of this as there are technical differences.

Again, we are surprised that this issue is raised at this point, because both datasets (MSL2tg ChIP-seq in Figure 1 and ChIP-qPCR in Suppl. Figure 2i) were already provided in the first submission of our manuscript.

In the text we state: “We performed ChIP- qPCR experiments and confirmed the binding of MSL2-HA to *vg*, *ap* and *en* at levels comparable to HAS, whilst two autosomal controls (*ent2* and *CG15011* promoter) were not enriched (Supplementary Fig. 2i).”

Using this phrasing, our intention was to relate to the fact that autosomal sites and HAS are similarly enriched in MSL2-HA ChIP-qPCR. Of course, we cannot compare these enrichments to MSL2tg FLAG ChIP-seq, because - as pointed out by the reviewer - there are significant technical differences:

- Figure 1b represents enrichments scored from ChIP-seq (genome browser snapshots). Here, the y-scale represents the difference of sequencing depth from normalized ChIP versus Input. Since the normalization scales the total coverage to be 1, a difference of 0 means that both ChIP and Input had about the same number of normalized reads, while values over zero mean an enrichment of ChIP over Input and negative values mean an enrichment of Input over ChIP (=background). This ChIP has been performed with a FLAG-tag antibody; the epitope is DYKDDDDK.
- The Suppl. Fig. 2i enrichments are determined by qPCR and the ChIP was performed using the HA tag (epitope YPYDVPDYA). We have again further commented on this figure for Reviewer #1 and copy this reply again below. It is important to note that the enrichments in ChIP-qPCR reported by us are consistent with what has been previously published by other labs.

Our earlier response to Reviewer #1 regarding ChIP-qPCR:

In the experiment displayed in Suppl. Fig. 2d, we have chosen the same y-axis scaling as Straub et al. (2013), where Figures 1D and 1E show a roughly 5 to 10 fold enrichment of MSLs over non-targets. Prestel et al. report a roughly 4-fold enrichment of MSL2 on the X-linked *armadillo* over non-targets (Figure 2E, Mol Cell 2010). We are also aware of MSL2 ChIP data published by the Larsson (Figueiredo et al., PLoS Genetics 2014) and Becker labs (Straub et al., PLoS Genetics 2008), but both do not provide a qPCR quantification for MSL2. The MSL2 ChIP in Figure 5A of Larschan et al. (Mol Cell, 2007) is presented as “% IP for MSL2 normalized to Input and PKA” and bars for HAS reach to 350%. This particular scaling is unclear to us. In Larschan et al. (PLOS Genetics, 2012) MSL2 ChIP enrichments are reported by setting the wild-type enrichment to 100%.

A question previously raised was whether autosomal sites identified in the present study using ChIP overlap with those previously mapped cytologically. This question was not addressed. It would be useful to know how closely molecular and cytological approaches harmonize. This requires simply consulting the literature.

The concern from Reviewer #2 raised after review of our initial manuscript was:

“Autosomal MSL binding sites have been previously studied. No effort is made to determine if sites currently identified correspond to ones mapped previously.”

Given the resolution and detection limits of microscopy, we assumed that Reviewer #2 was referring to previously published MSL ChIP-seq data. As indicated in the point-by-point response, we had provided further analyses and comparisons with previously

published data in the revised version of our manuscript. For example, some autosomal sites overlap with CLAMP, MSL1, roX or H4K16ac (Figure 2).

As explained in the point-by-point response and our manuscript, we think that overlaps with immunostainings from salivary glands, which do not express such developmental regulatory genes (Figure 2g), might be misleading. Moreover, cytological sites correspond to several dozens of genes and thereby provide very limited and imprecise mapping information (e.g. Vatolina et al., PLOS One 2011; V. A. Khoroshko, 2018). For these reasons, we prefer to correlate our ChIP-seq data to existing ChIP-seq profiles published in the last 5 years, rather than correlating them to 15-year-old microscopy data.

To specifically comment on the autosomal MSL2 sites detected by Demakova et al. (2003): In this publication, autosomal sites were observed in a stochastic fashion and under very specific conditions: Demakova et al. expressed both *msl-1* and *msl-2* in males, which leads to very high expression of MSL2 (see the Western provided in Figure 1a, lane 7 in Demakova et al.). Please also note the nature of this particular *hsp83-msl-2* line, which was originally reported by Kelley et al. (Cell, 1995). This transgene contains two closely linked insertions of *msl-2* (reported in Lyman et al., Genetics 1997) and it triggers almost complete female-specific lethality, that we and others (Hudry et al., Nature, 2016) do not observe with our transgenes. Chang & Kuroda (1998) report that upon ectopic expression of *msl-1*, any surviving females in *hsp83-msl-2* are completely killed, which relates to even higher expression levels than observed with *hsp83-msl-2* alone. Conversely, expression of *msl-2* using the endogenous promoter (NOPU-MSL2) does not trigger toxic effects in females (Lim & Kelley, PLOS Genetics, 2012). Therefore, it can be concluded that in the conditions used by Demakova et al. – in contrast to our conditions – unusually high overexpression levels of MSL2 have been achieved.

Demakova et al. indeed note on p.110: “Oregon R male. Few sites are detected on autosomes.” and “Multiple autosomal sites are detected under conditions of overexpression of both MSL1 and MSL2.” and further “Autosomal sites are not detected under reduced MSL2 levels. The only visible site is detected at region 67D in each nucleus.” (The 67D region corresponds to the insertion site of the *msl-2* transgene itself). Unfortunately, there is no table in the publication that reports the Oregon R results separately.

Given the very different biological and experimental setups of these data, we have kept the current displays in Figures 1 and 2 and Supplementary Figures 1 and 2, in which we consistently compare high-resolution ChIP/ChIRP-seq data. In order to make it possible for the interested reader to easily correlate our data to stainings from salivary glands, we have now added the respective cytological positions of the selected example genes shown in genome browser snapshots in the Figure legends.

A reviewer asked about the presence of CLAMP at autosomal sites. The authors provided documentation of CLAMP at X-linked sites, something that is already well-known, but failed to address the presence or absence of CLAMP at autosomal sites.

The concern from Reviewer #2 raised after review of our initial manuscript was:

Do the autosomal fly HAS identified in this study bind CLAMP?

Figure 2a shows exactly this experiment and we had already answered this request in the point-by-point response to Reviewer #2. Cluster 1 autosomal sites (GAGA-rich motif) are indeed enriched for CLAMP.